# Top-down descending facilitation of spinal sensory excitatory transmission from the anterior cingulate cortex

Tao Chen[1,2], Wataru Taniguchi[3,7], Qi-Yu Chen[1], Hidetoshi Tozaki-Saitoh (ORCID) [4], Qian Song[1], Ren-Hao Liu[1], Kohei Koga[1,5], Tsuyoshi Matsuda[4], Yae Kaito-Sugimura[3], Jian Wang[2], Zhi-Hua Li[2], Ya-Cheng Lu[2], Kazuhide Inoue[5], Makoto Tsuda[4], Yun-Qing Li[2], Terumasa Nakatsuka[3] & Min Zhuo[1,6]

Spinal sensory transmission is under descending biphasic modulation, and descending facilitation is believed to contribute to chronic pain. Descending modulation from the brainstem rostral ventromedial medulla (RVM) has been the most studied, whereas little is known about direct corticospinal modulation. Here, we found that stimulation in the anterior cingulate cortex (ACC) potentiated spinal excitatory synaptic transmission and this modulation is independent of the RVM. Peripheral nerve injury enhanced the spinal synaptic transmission and occluded the ACC-spinal cord facilitation. Inhibition of ACC reduced the enhanced spinal synaptic transmission caused by nerve injury. Finally, using optogenetics, we showed that selective activation of ACC-spinal cord projecting neurons caused behavioral pain sensitization, while inhibiting the projection induced analgesic effects. Our results provide strong evidence that ACC stimulation facilitates spinal sensory excitatory transmission by a RVM-independent manner, and that such top-down facilitation may contribute to the process of chronic neuropathic pain.

[1] Center for Neuron and Disease, Frontier Institute of Science and Technology, Xi'an Jiaotong University, Xi'an 710054, China. [2] Department of Anatomy, Histology and Embryology and K.K. Leung Brain Research Centre, The Fourth Military Medical University, 710032 Xi'an, China. [3] Pain Research Center, Kansai University of Health Sciences, Kumatori, Osaka 590-0482, Japan. [4] Department of Life Innovation, Graduate School of Pharmaceutical Sciences, Kyushu University, Fukuoka 812-8582, Japan. [5] Department of Molecular and System Pharmacology, Graduate School of Pharmaceutical Sciences, Kyushu University, Fukuoka 812-8582, Japan. [6] Department of Physiology, Faculty of Medicine, Center for the Study of Pain, University of Toronto, 1 King's College Circle, Toronto, ON M5S 1A8, Canada. [7] Present address: Department of Orthopaedic Surgery, Wakayama Medical University, Wakayama 641-8510, Japan. These authors contributed equally: Tao Chen, Wataru Taniguchi, Qi-Yu Chen. Correspondence and requests for materials should be addressed to Y.-Q.L. (email: deptanat@fmmu.edu.cn) or to T.N. (email: nakatsuka@kansai.ac.jp) or to M.Z. (email: minzhuo@utoronto.ca)

It is well documented that spinal sensory transmission including nociceptive transmission is under descending biphasic modulation from supraspinal structures[1–4]. Electrical or chemical stimulation of the rostral ventromedial medulla (RVM) in brainstem, produces both inhibitory and facilitatory effects of spinal dorsal neuron responses and spinal nociceptive reflexes to noxious stimuli[5–7]. In pathological pain conditions, descending facilitatory system has been reported to be activated or enhanced[2,8–13]. However, most of previous studies focus on descending modulation from the brainstem, there is no direct evidence that cortex may regulate spinal nociceptive transmission at neuronal level.

Anterior cingulate cortex (ACC) is an important cortical area for pain perception, chronic pain, and its related emotional disorders[14–17]. Activation of ACC has been reported to induce pain affected aversive learning and fear[18–20], while inhibition of ACC activity reduced chronic pain and pain-related unpleasant responses[15,16,21,22]. Clinically, lesions of ACC significantly reduce chronic pain-related suffering[22]. Interestingly, stimulation of ACC has been reported to cause facilitation of a spinal nociceptive reflex[23]. A recent study found that nerve injury caused potentiation of postsynaptic responses of ACC deep layer neurons that send direct projection to the spinal cord dorsal horn[24]. However, it is unclear if ACC neurons may regulate spinal nociceptive sensory transmission through this direct ACC-spinal-projecting system. Furthermore, facilitation of behavioral withdrawal reflexes may be indirectly caused by alteration of the activity of motor neurons[25] or glia cells[26] in the spinal cord. It is critical to directly confirm ACC stimulation may indeed facilitate spinal sensory synaptic transmission in whole animal preparations.

In the present study, by using in vivo whole-cell patch recording and $Ca^{2+}$ imaging works on the spinal dorsal horn (SDH) neurons, we found that electric stimulation of ACC directly potentiated the spinal excitatory synaptic transmission and spinal neuronal $Ca^{2+}$ responses. After peripheral nerve injury, spinal excitatory synaptic transmission was enhanced and the ACC-spinal top-down facilitation was occluded. By combining retrograde/anterograde labeling and electron microscopic (EM) methods, we confirmed that ACC-spinal cord projecting fibers made excitatory synapses with SDH neurons, especially the projecting neurons and excitatory interneurons. Finally, by employing optogenetic technique, we found that selectively activation of ACC-spinal cord projecting neurons induced behavioral sensitization in pain responses, while inhibition of them alleviated the neuropathic pain. Our findings thus show that ACC directly potentiates spinal sensory transmission and the top-down facilitation contributes to the process of chronic neuropathic pain.

## Results

**ACC stimulation potentiated the sEPSCs of SDH neurons.** In order to check whether activation of ACC could affect the synaptic transmission of spinal cord neurons, we performed in vivo whole-cell patch-clamp recording of spinal laminae I/II neurons that located at a depth less than 150 µm of the lumber SDH from naïve adult rats. We first recorded the spontaneous excitatory postsynaptic currents (sEPSCs) from 12 spinal neurons. At the spinal cord level, spontaneous activities of SDH neurons have been linked to pain, especially in pathological conditions[27–30]. After obtaining stable recording of sEPSCs (under voltage-clamp conditions at a $V_H$ of −70 mV, all neurons exhibited sEPSCs with a mean frequency of 7.9 ± 1.1 Hz and a mean amplitude of 15.4 ± 2.0 pA), we applied focal electrical stimulation (100 Hz at the intensity of 100 µA) to the contralateral ACC through implanted electrodes. Stimulation of

the ACC potentiated both the frequency (126.2 ± 7.8% of the baseline, $p < 0.05$) and amplitude (120.3 ± 7.3% of the baseline, $p < 0.05$) of the spinal sEPSCs (Fig. 1), suggesting that ACC activation may trigger top-down facilitation of spinal excitatory sensory transmission.

Spinal inhibitory postsynaptic currents (sIPSCs) are important mechanism for inhibitory control of sensory information in the spinal cord[25,26]. We then wanted to determine whether ACC stimulation affect spinal inhibitory transmission. sIPSCs were recorded from 7 SDH neurons, and the same ACC stimulation did not cause any change of sIPSCs frequency (mean 96.9 ± 2.0% of the baseline) or amplitude (96.0 ± 1.8% of the baseline) (Supplementary Fig. 1).

**Occlusion of ACC top-down facilitation in neuropathic pain.** Long-term potentiation (LTP) along somatosensory pathways is a key mechanism for the central sensitization of chronic pain[9,31,32]. It has been shown that, in different animal models of chronic pain including neuropathic pain, sensory synaptic transmission[33] and spontaneous EPSCs[27,34] of SDH neurons are significantly potentiated. We then compared the sEPSC from rats receiving sham surgery or 7 days after spared nerve injury (SNI) (Fig. 2a). In sham surgery rats, the frequency and amplitude of the sEPSC were similar to those in naïve rats (frequency: 9.0 ± 1.3 Hz, amplitude: 12.3 ± 0.8 pA, $n = 20$, $p > 0.05$ compared with naïve group). However, in rats with neuropathic pain (Fig. 2b) after SNI surgery, both the frequency and amplitude of the sEPSC were greatly potentiated (frequency: 15.9 ± 1.6 Hz, amplitude: 21.8 ± 3.1 pA, $n = 20$, $p < 0.01$ compared with sham group) (Fig. 2c-d).

Previous studies have consistently suggested that descending facilitation contributes to spinal potentiation of pain transmission (see Introduction). We thus wanted to examine if the ACC-spinal top-down facilitation may be activated in chronic pain condition. We expect that direct ACC stimulation-induced facilitatory effect of the spinal sEPSCs will be reduced or occluded if the system has been activated. To test this, we repeated experiments of ACC stimulation in animals with sham treatment or nerve injury. In rats with SNI, we found that ACC stimulation did not induce any significant potentiation of either frequency (98.2 ± 8.6% of the baseline, $n = 9$, $p > 0.05$) or amplitude (105.1 ± 5.2% of the baseline, $n = 9$, $p > 0.05$) of the sEPSCs (Fig. 2e–g). By contrast, in rats with sham operation, stimulation in the ACC produced significant potentiation of the frequency (144.2 ± 14.6% of the baseline, $n = 8$, $p < 0.05$) and amplitude (110.7 ± 2.7% of the baseline, $n = 8$, $p < 0.01$) (Fig. 2g), similar to the results from naïve rats. These results suggest that the descending facilitation may contribute to the potentiated sEPSCs after nerve injury.

**RVM is not required for ACC top-down facilitation.** Previous studies have shown that the RVM is important for the descending pain regulation, as well as the responses of SDH neurons to peripheral noxious stimuli[1,4,35]. To test if the RVM is required for the ACC-induced top-down facilitation, we injected lidocaine (2%, 0.5 µl) or CNQX (10 mM, 0.5 µl) into the RVM to block its neuronal activity and synaptic transmission before or after ACC stimulation.

We firstly performed RVM blockade with lidocaine injection before ACC stimulation in sham surgery rats ($n = 7$). It is found that, after lidocaine injection, the frequency of the spinal sEPSC was increased in four neurons and decreased in three neurons, while the amplitude of the sEPSC was increased in one neuron, decreased in five neurons, and not changed in one neuron (Fig. 3). However, following ACC stimulation subsequentially potentiated the frequency and amplitude at almost all neurons

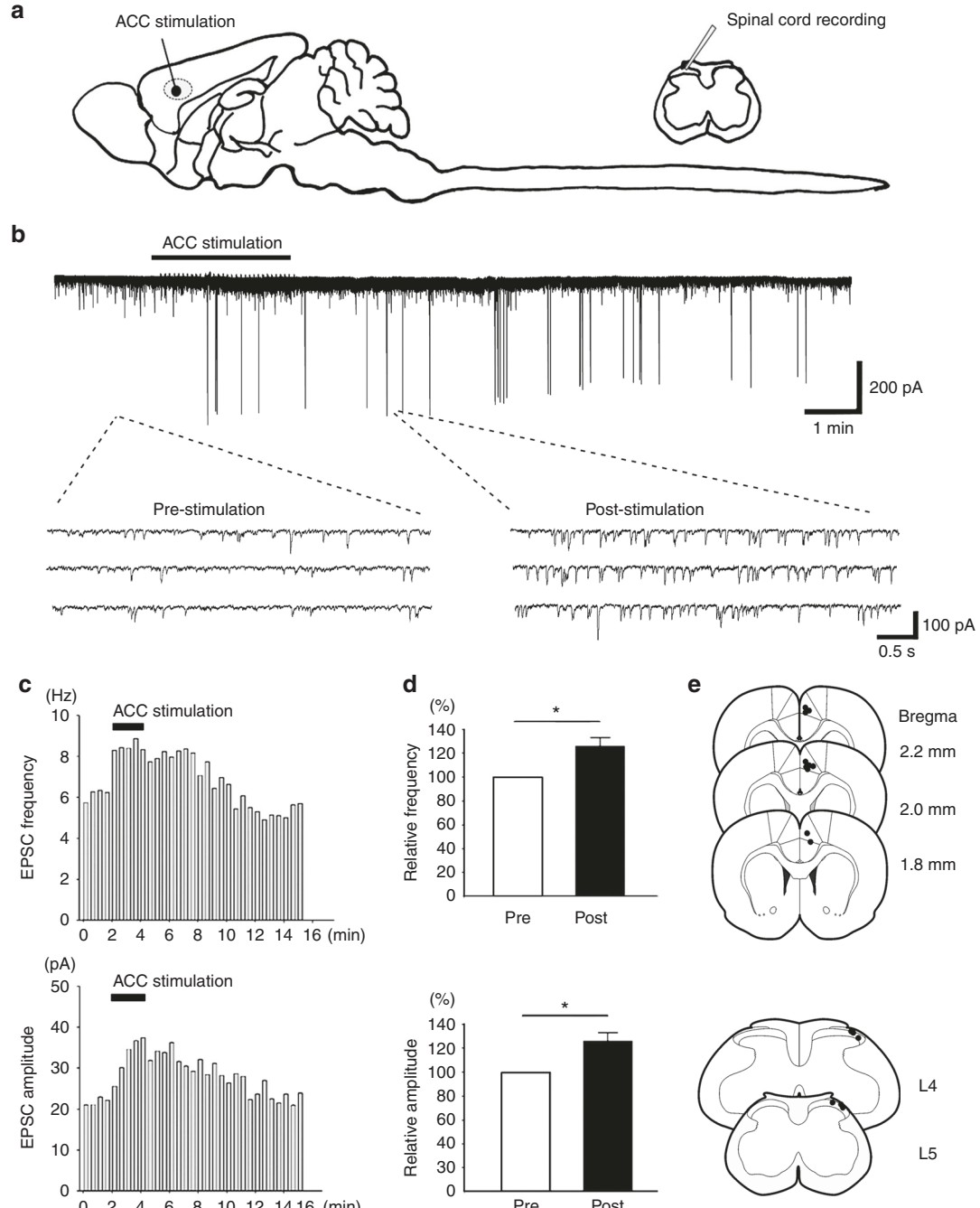

**Fig. 1** ACC stimulation-induced facilitation of the sEPSCs on the SDH neurons. **a** A schematic diagram showing the experimental design for ACC electronic stimulation and in vivo spinal cord recording. **b–c** One sample (**b**) and histogram figure (**c**) showing ACC stimulation-induced facilitation of the frequency and amplitude of the sEPSC. **d** Summarized results from 12 SDH neurons. *frequency: $p = 0.01$; amplitude: $p = 0.03$, paired $t$-test. **e** Mapping of the stimulation sites in the ACC and locations of recorded SDH neurons in the lumber spinal cord

(lidocaine: frequency: $110.6 \pm 37.9\%$ of the baseline, amplitude: $83.3 \pm 9.3\%$ of the baseline, $p > 0.05$ in comparison with the baseline; ACC stimulation: frequency: $135.1 \pm 39.8\%$ of the baseline, amplitude: $96.1 \pm 8.9\%$ of the baseline, $p < 0.05$ in comparison with the post-lidocaine injection). Similar results could be seen in case of CNQX injection ($n = 6$). ACC stimulation potentiated both the frequency and amplitude of the sEPSC under the condition of CNQX blockade (CNQX: frequency: $81.4 \pm 7.2\%$ of the baseline, $p < 0.05$ in comparison with the baseline, amplitude: $87.9 \pm 6.9\%$ of the baseline, $p > 0.05$ in comparison with the baseline; ACC stimulation: frequency:

$93.2 \pm 8.3\%$ of the baseline, amplitude: $102.0 \pm 5.8\%$ of the baseline, $p < 0.05$ in comparison with the post-CNQX injection).

We then checked whether ACC stimulation-induced potentiation of spinal sEPSC could be reversed by following RVM injection of lidocaine in sham rats ($n = 8$) (Fig. 4). It is found that ACC stimulation potentiated the frequency and the amplitude of the spinal sEPSC, which was not significantly changed by lidocaine injection (ACC stimulation: frequency: $135.6 \pm 14.7\%$ of the baseline, amplitude: $124.2 \pm 10.3\%$ of the baseline, $p < 0.05$ in comparison with the baseline; lidocaine: frequency: $137.8 \pm 19.9\%$ of the baseline, amplitude: $123.3 \pm 9.8\%$ of the

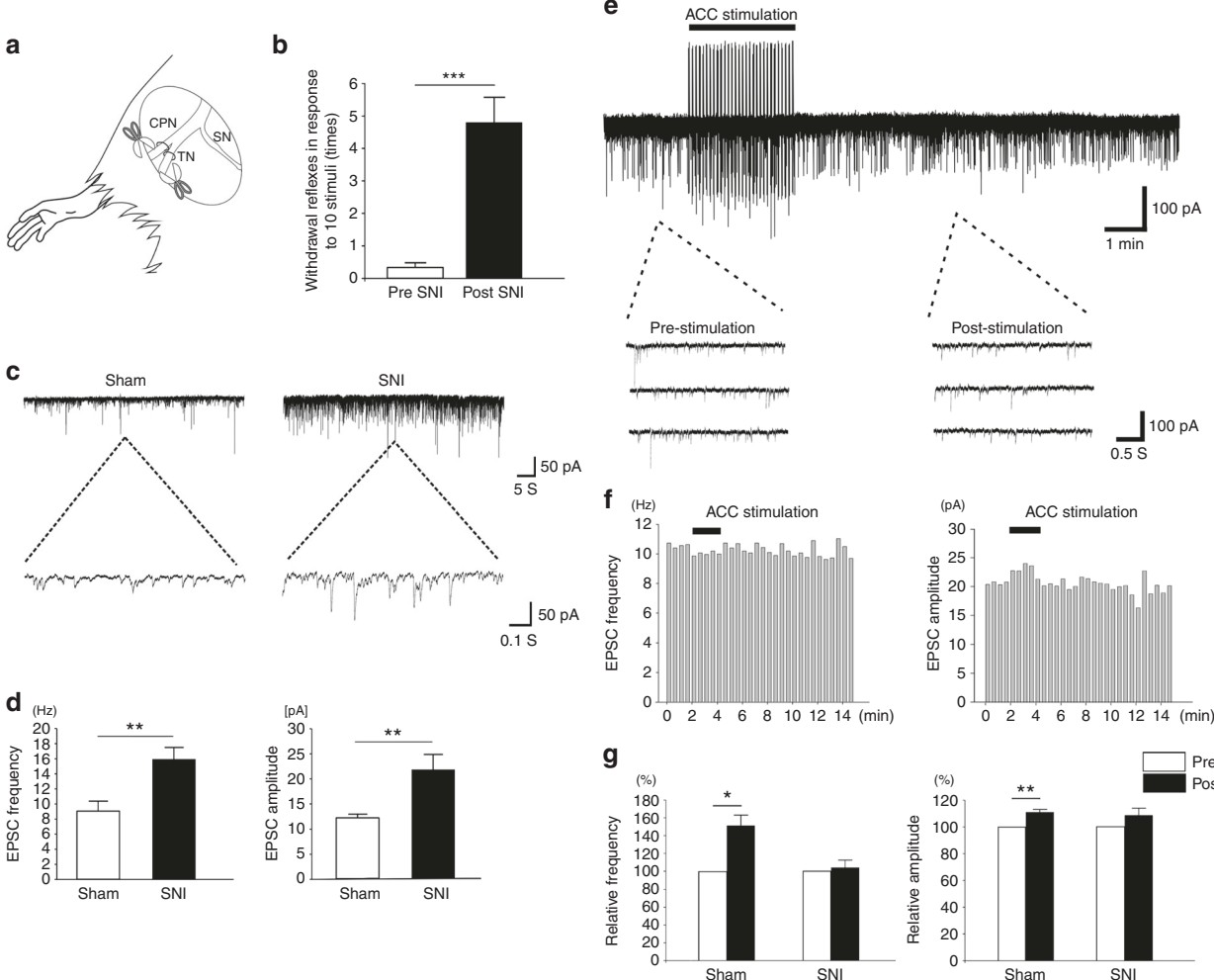

**Fig. 2** Potentiated sEPSCs of SDH neurons and the occlusion of ACC-induced facilitation in rats with neuropathic pain. **a** A schematic diagram showing the SNI model. The tibial nerve (TN) and common peroneal nerve (CPN) were cut and ligated but leaving the sural nerve (SN) intact. **b** A significant mechanical allodynia was observed in rats 7 days after SNI surgery (pre SNI, 0.36 ± 0.25 times, post SNI, 4.82 ± 0.78 times, $n = 11$, ***$p < 0.001$, unpaired $t$-test). **c** Recording samples showing the frequency and amplitude of the sEPSCs of spinal SDH neurons in SNI rats were significantly potentiated than those in sham operated rats. **d** Summarized results from 20 SDH neurons in sham and SNI group, respectively. **frequency: $p = 0.002$; amplitude: $p = 0.005$, unpaired $t$-test. **e** One recording sample showing ACC electrical stimulation could not facilitate the sEPSCs of 1 SDH neuron in rats with SNI. **f** Histogram figures showing the frequency and amplitude of the sample sEPSCs in **e** before and after ACC stimulation. **g** Summarized results from 8 SDH neurons in sham group and nine neurons in SNI group. *frequency: $p = 0.01$; amplitude: $p = 0.002$, paired $t$-test

baseline, $p > 0.05$ in comparison with the post-ACC stimulation). Similar results could be seen with CNQX injection ($n = 7$) (ACC stimulation: frequency: 131.3 ± 7.7% of the baseline, amplitude: 116.5 ± 4.7% of the baseline, $p < 0.05$ in comparison with the baseline. CNQX: frequency: 131.6 ± 14.0% of the baseline, amplitude: 118.2 ± 6.8% of the baseline, $p > 0.05$ in comparison with the post-ACC stimulation). In sum, these results suggest that RVM is not required for the facilitatory effect on spinal sEPSCs induced by ACC stimulation.

We then feel interested to know whether the combined RVM blockade and ACC stimulation could affect the spinal sEPSC in SNI rats (Fig. 5). Interestingly, similar with the sham group, lidocaine injection into the RVM did not change the frequency and amplitude of the spinal sEPSC on the whole (frequency: 97.0 ± 6.8% of the baseline, amplitude: 94.0 ± 2.4% of the baseline, $p > 0.05$ in comparison with the baseline). However, following ACC stimulation did not affect the sEPSC either (frequency: 95.1 ± 16.6% of the baseline, amplitude: 86.3 ± 4.1% of the baseline, $p > 0.05$ in comparison with the post-lidocaine injection). Furthermore, ACC stimulation with followed lidocaine injection

did not change the spinal sEPSC (ACC stimulation: frequency: 111.4 ± 29.1% of the baseline, amplitude: 92.3 ± 4.5% of the baseline, $p > 0.05$ in comparison with the baseline; lidocaine: frequency: 121.2 ± 35.5% of the baseline, amplitude: 87.0 ± 2.0% of the baseline, $p > 0.05$ in comparison with the post-ACC stimulation).

**Fos expression in the SDH after ACC stimulation.** Fos protein is a widely used marker for detecting the neuronal activity in SDH[36,37]. We tested whether the Fos expression of the SDH should be altered by ACC stimulation. Since ACC mainly send projection to the laminae I–II of the contralateral SDH[24,38] (also see Supplementary Fig. 3), we proposed that ACC stimulation should affect the Fos expression on superficial laminae of contralateral SDH. We then applied high-frequency stimulation on the right side of ACC and checked the Fos expression on the dorsal horn (laminae I–V) of lumbers cord 4–5 of rats ($n = 18$ slices/3 rats in each groups). In accordance with the electrophysiological results, we found that ACC stimulation selectively increased the Fos expression on the laminae I–II of left

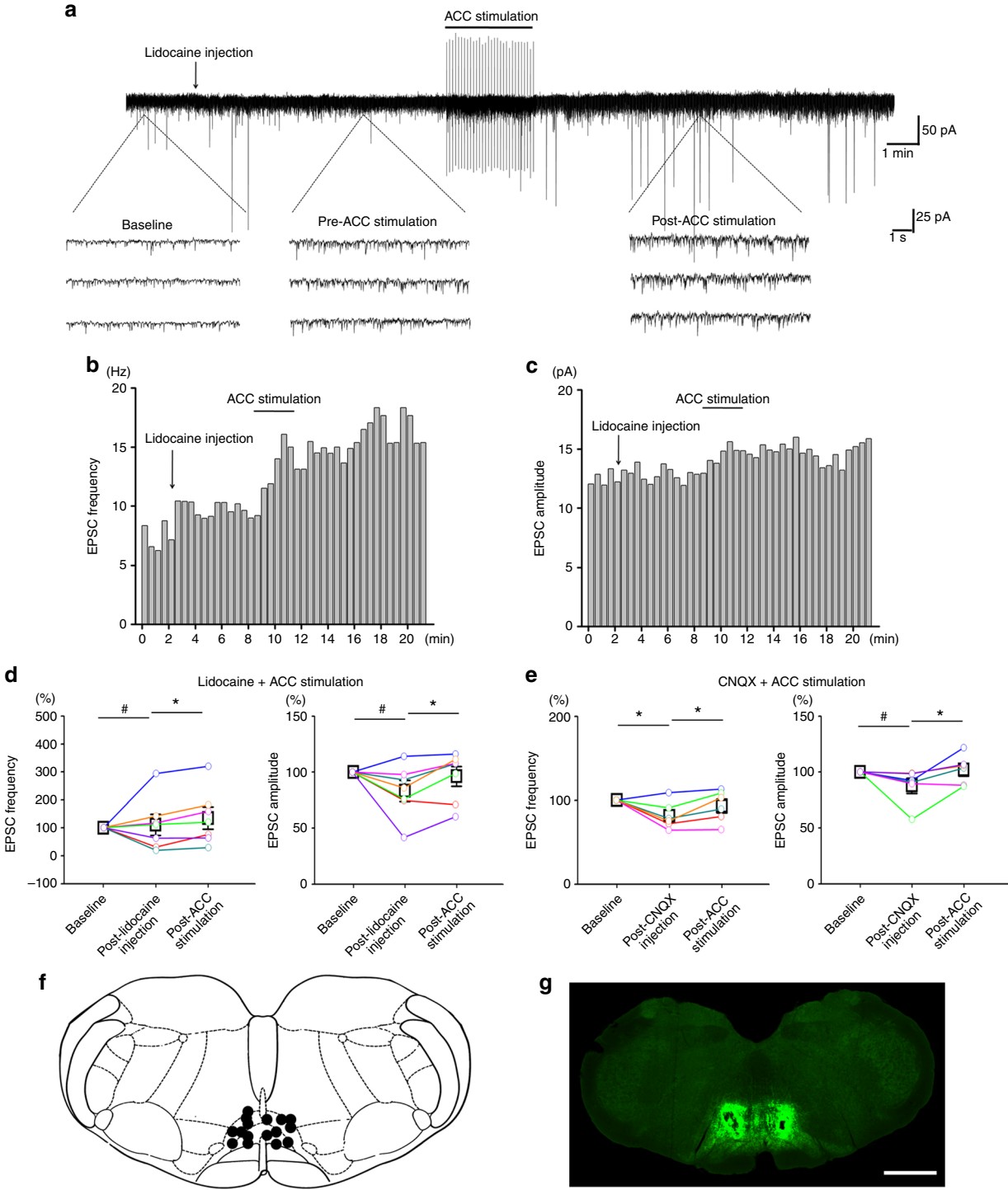

**Fig. 3** RVM blockade did not block ACC stimulation-induced potentiation of the spinal sEPSC in sham operated rats. **a–c** One sample (**a**) and the histogram figures showing the frequency (**b**) and amplitude (**c**) of the spinal sEPSC with lidocaine blockade of RVM were potentiated by following ACC stimulation. **d** Summarized results from 7 SDH neurons with lidocaine injection and ACC stimulation. *frequency: $p = 0.01$; amplitude: $p = 0.02$. One-way RM ANOVA. **e** Summarized results from 6 SDH neurons with CNQX injection and ACC stimulation. *frequency: CNQX injection vs. baseline, $p = 0.04$, ACC stimuli vs. CNQX injection, $p = 0.03$; amplitude: $p = 0.04$. One-way RM ANOVA. **f–g** Schematic diagram and sample figures showing the lidocaine injection sites on the RVM. #$p > 0.05$. Bar = 1 mm in **g**

($p < 0.05$) but not right ($p > 0.05$) SDH in sham operated rats. ACC stimulation did not increased the Fos expression on deep laminae (III–V) either (Supplementary Figs. 2a,b and i). Interestingly, RVM lidocaine injection did not affect the Fos expression in laminae I–III ($p > 0.05$) but increased the Fos expression ($p < 0.05$) in laminae IV–V of both sides of SDH (Supplementary

Fig. 2c and i). In rats with RVM lidocaine injection, following ACC stimulation could still increase the Fos expression in laminae I–II ($p < 0.05$) but not in laminae III–V ($p > 0.05$) (Supplementary Fig. 2d and i).

In rats one week after SNI, Fos expression was significantly increased in laminae I–II and IV–V of both sides of the SDH

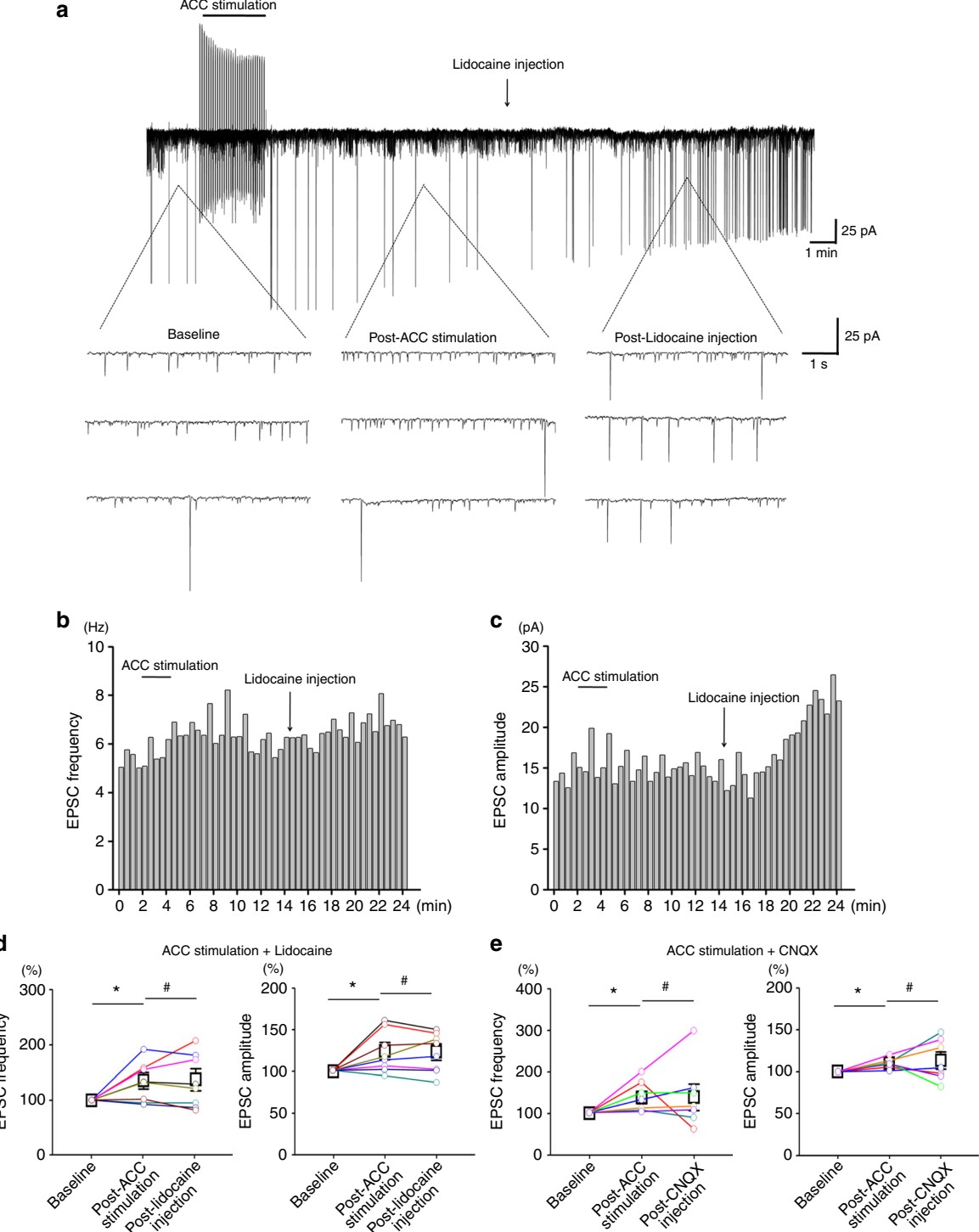

**Fig. 4** RVM blockade did not reverse ACC stimulation-induced potentiation of the spinal sEPSC in sham operated rats. **a–c** One sample (**a**) and the histogram figures showing the frequency (**b**) and amplitude (**c**) of the spinal sEPSC were potentiated by ACC stimulation, which could not be reversed by following lidocaine blockade of RVM. **d** Summarized results from 8 SDH neurons with ACC stimulation and lidocaine injection. *frequency: $p = 0.03$; amplitude: $p = 0.04$. One-way RM ANOVA. **e** Summarized results from 7 SDH neurons with ACC stimulation and CNQX injection. *frequency: $p = 0.04$; amplitude: $p = 0.02$. One-way RM ANOVA. #$p > 0.05$

($p < 0.05$) (Supplementary Fig. 2e, i and j). ACC stimulation or lidocaine injection could not increase the Fos expression in laminae I–V ($p > 0.05$) (Supplementary Fig. 2f, g and j). In rats with RVM lidocaine injection and following ACC stimulation, the Fos expression was not changed ($p > 0.05$) (Supplementary Fig. 2h and j).

**ACC inhibition reduced spinal sEPSC in SNI rats**. Since the ACC-spinal cord facilitation may contribute to the potentiated spinal sEPSCs in nerve injury condition, we consider that inhibition of the top-down potentiation would sequentially inhibit the enhanced sEPSC. In our previous works, we have shown that peripheral injury induces LTP in ACC-spinal cord projecting

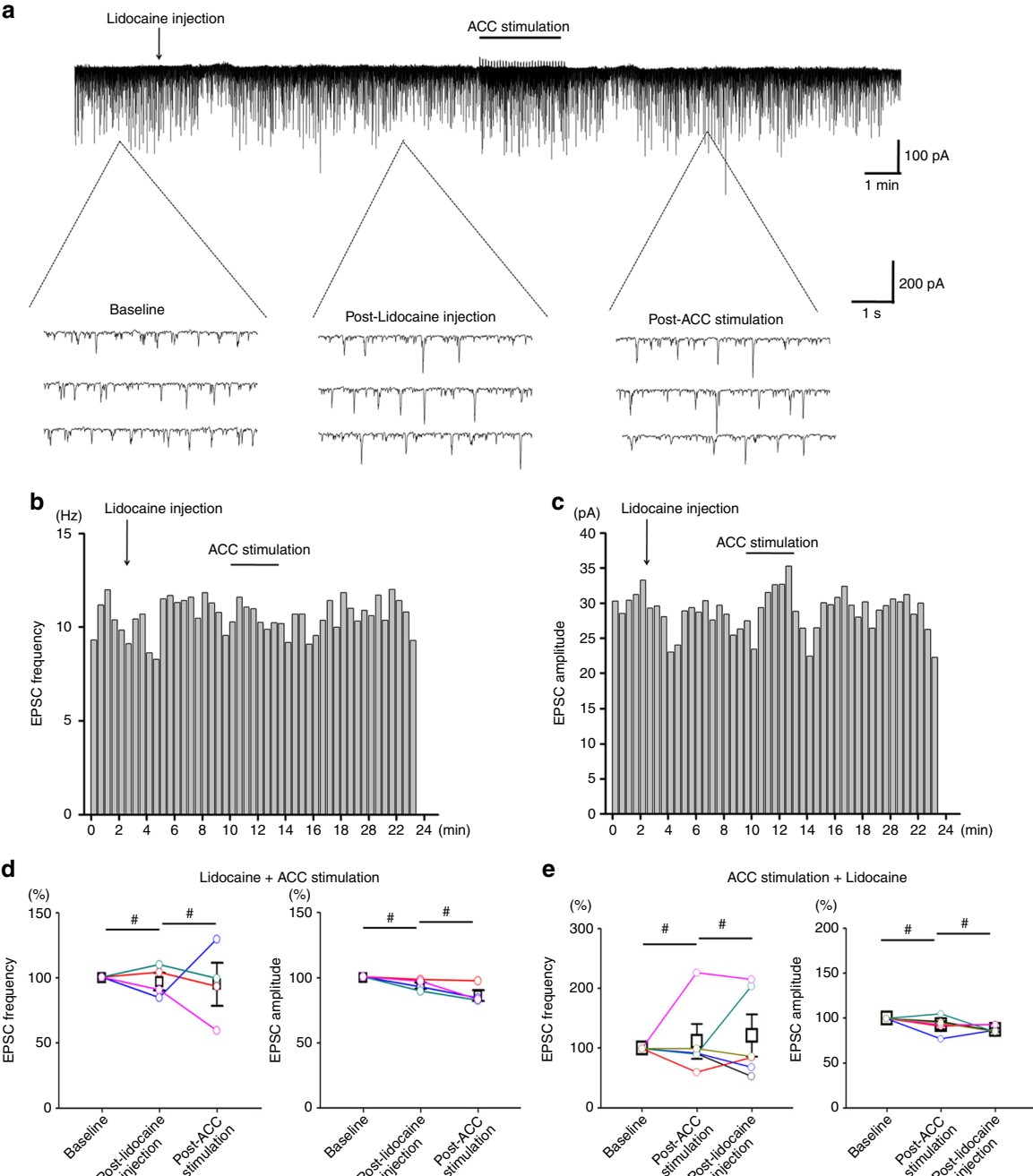

**Fig. 5** RVM blockade and ACC stimulation did not affect the spinal sEPSC in SNI rats. **a–c** One sample (**a**) and the histogram figures showing the frequency (**b**) and amplitude (**c**) of the spinal sEPSC was not affected by lidocaine blockade of RVM and following ACC stimulation. **d** Summarized results from 4 SDH neurons with lidocaine injection and ACC stimulation. **e** Summarized results from 4 SDH neurons with ACC stimulation and following lidocaine injection. #$p > 0.05$

neurons, which is mainly mediated by the increased post-synaptic expression of AMPA receptor and can be erased by a selective AMPA receptor antagonist NAPSM[24,38]. We expect that the AMPA receptor mediated LTP may contribute to the ACC-spinal cord descending facilitation in chronic pain condition. To test this possibility, we locally applied NASPM (5 mM, 0.5 µl) into the ACC, and examine the effect on sEPSCs of SDH neurons. In sham operated rats, NASPM application produced no effects on the frequency or amplitude of the sEPSCs (frequency: $90.6 \pm 8.2\%$ of baseline, $p > 0.05$; amplitude: $99.5 \pm 8.6\%$ of baseline, $p > 0.05$, $n = 10$) (Fig. 6a–c). However, in SNI rats, NASPM application

significantly inhibited the frequency but not the amplitude of the sEPSCs (frequency: $68.6 \pm 9.3\%$ of baseline, $p < 0.05$; amplitude: $103.3 \pm 13.4\%$ of baseline, $p > 0.05$, $n = 6$) (Fig. 6d–f). These results further support that, in rats with nerve injury, the potentiated sEPSC in spinal cord may due to the enhanced ACC descending inputs. Because NASPM microinjection into the ACC has been reported to produce analgesic effects in chronic pain[39], the enhanced ACC-spinal cord descending facilitation likely contributes to the behavioral sensitization in chronic pain conditions.

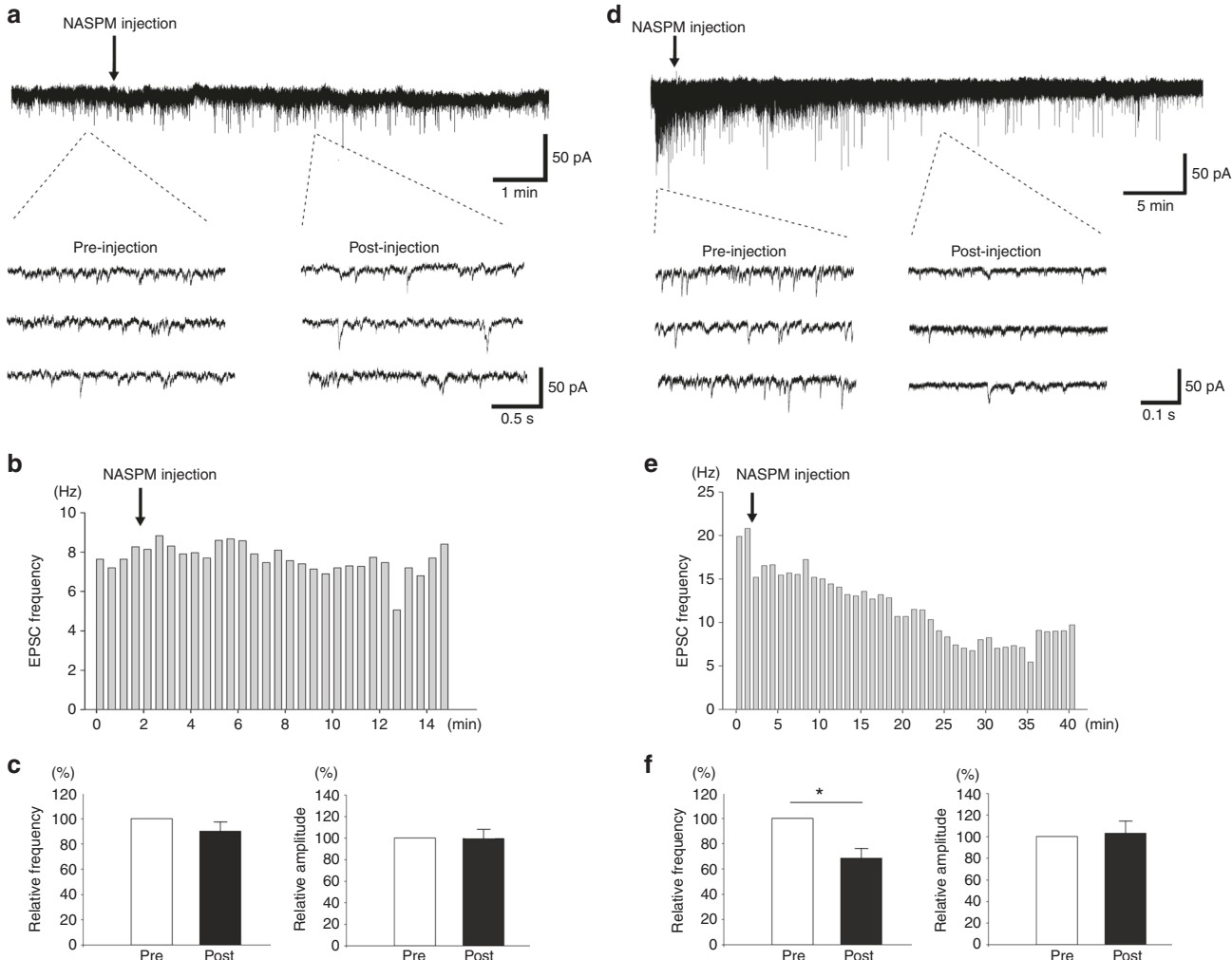

**Fig. 6** ACC inhibition alleviated sEPSCs of SDH neurons in rats with SNI. **a** One recording sample showing ACC injection of NASPM didn't change the basal sEPSCs of the SDH neurons in rats with sham surgery. **b** Histogram figure showing the frequency of the sample sEPSCs in **a** before and after NASPM injection. **c** Summarized results from 10 SDH neurons in sham rats. **d** One recording sample showing ACC injection of NASPM significantly inhibited the frequency but not the amplitude of the sEPSCs of spinal SDH neurons from rats with SNI. **e** Histogram figure shows the frequency of the sEPSC of 1 SDH neuron before and after NASPM injection. **f** Summarized results from 6 SDH neurons in SNI rats. *$p = 0.03$, paired $t$-test

**ACC stimulation potentiated sensory synaptic responses**. We then wanted to examine if ACC stimulation affect the synaptic inputs to SDH neurons by peripheral sensory stimulation as well. Innocuous (air puffs) or noxious (pinch) mechanical stimuli were applied to the ipsilateral receptive field of SDH neurons in sham operated or SNI rats. Either air puff or pinch stimuli evoked a barrage of EPSCs (eEPSCs), which disappeared within 1 s after stimuli were removed (Fig. 7). The frequency and the amplitude of the eEPSCs during the stimuli were analyzed and summarized. In rats with sham operation, the frequency and amplitude of eEPSCs during the puff stimuli was $20.8 \pm 1.5$ Hz and $23.6 \pm 5.9$ pA ($n = 9$). ACC electrical stimulation increased both the frequency and amplitude (frequency: $112.6 \pm 3.5\%$ of baseline, $p < 0.01$; amplitude: $170.6 \pm 31.1\%$ of baseline, $p < 0.05$) (Fig. 7a). During the pinch stimuli, the frequency and amplitude of eEPSCs was $29.7 \pm 2.0$ Hz and $33.2 \pm 5.8$ pA ($n = 9$) and ACC electrical stimulation enhanced them either (frequency: $112.7 \pm 5.0\%$ of baseline, $p < 0.05$; amplitude: $150.4 \pm 17.5\%$ of baseline, $p < 0.05$) (Fig. 7b). In rats with nerve injury, however, ACC stimulation failed to induce potentiation of the eEPSCs. After ACC stimulation, the frequency reached $94.1 \pm 5.5\%$ of baseline in puff stimuli and $101.6 \pm 4.5\%$ in pinch stimuli ($p > 0.05$, $n = 7$), while the

amplitude reached $101.7 \pm 1.9\%$ of baseline in puff stimuli and $101.3 \pm 11.1\%$ in pinch stimuli ($p > 0.05$, $n = 7$) (Fig. 7c–d).

**ACC stimulation potentiated the Ca$^{2+}$ responses in SDH**. We then wanted to confirm the affection of ACC stimulation to the neuronal activity in the SDH. For doing this, we transduced genetically encoded Ca$^{2+}$ indicator GCaMP6s to mice SDH neurons (L4-5) with AAV2/9 vector and tested their activity by employing in vivo two-photon Ca$^{2+}$ imaging works (Fig. 8a). Imaging depth was between 25–70 μm corresponding to laminae I-IIo. We found that basal spinal fluorescence was weak and little neuronal activities were observed without given paw stimulation. However, peripheral stimuli, especially pinch stimuli, could induce strong fluorescence, indicating the evoked neuronal Ca$^{2+}$ responses in the SDH (Fig. 8b). Of 80 identified GCaMP6s-expressing neurons, 72 (90%) neurons showed increased fluorescence immediately after nociceptive pinch stimulation and they were thus considered as responsive neurons. After ACC stimulation, the Ca$^{2+}$ activity from 54.2% of the responsive neurons was potentiated, from 19.4 to 26.4% of the responsive neurons was alleviated or unchanged (Fig. 8e) (Supplementary movie 1).

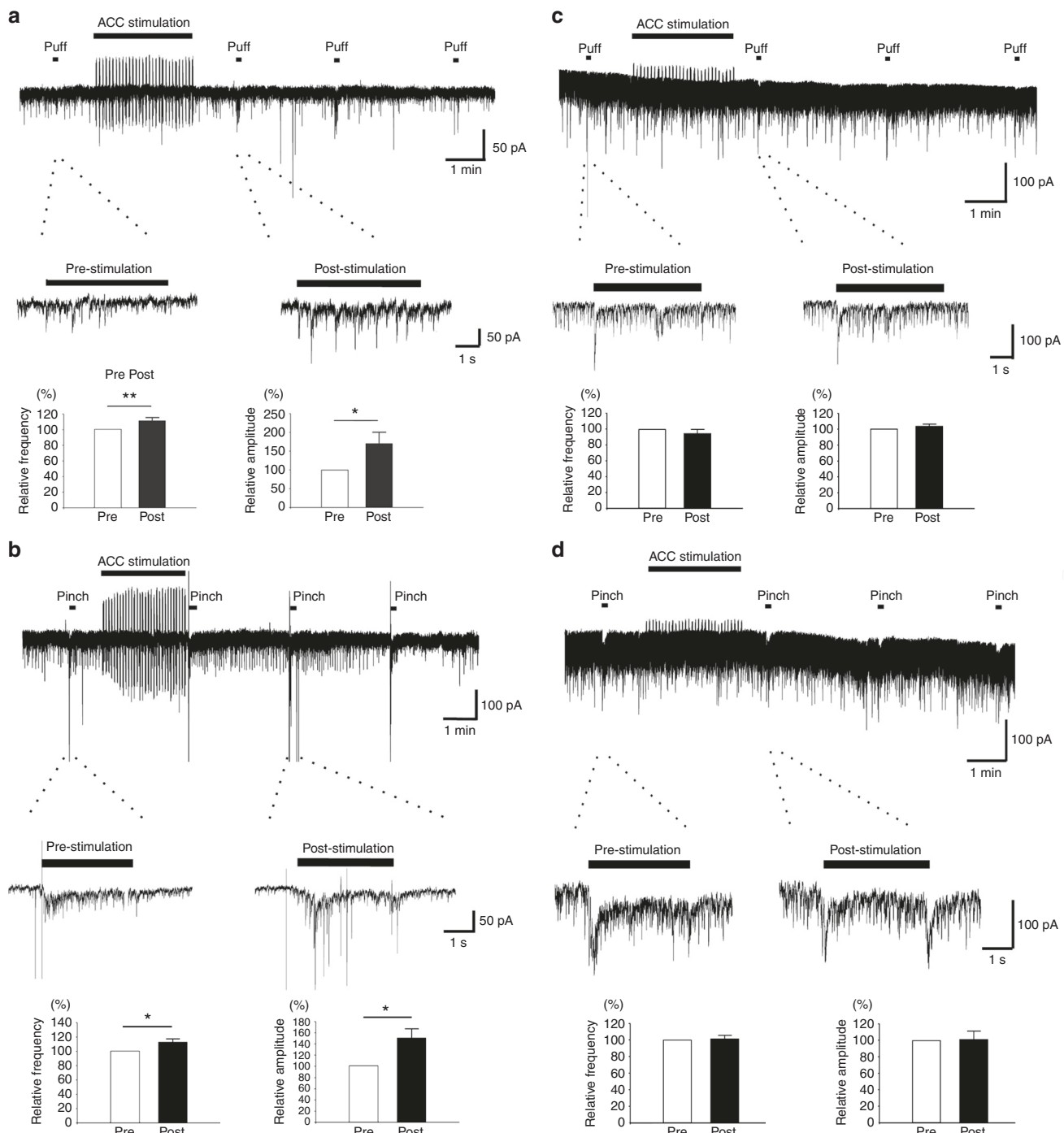

**Fig. 7** ACC stimulation potentiated the primary afferent stimulation induced EPSC of SDH neurons. **a** One recording sample and summarized results (9 SDH neurons) showing that ACC electrical stimulation increased the frequency and amplitude of puff induced EPSC. *$p = 0.04$; **$p = 0.006$, paired $t$-test. **b** One sample and summarized results (10 SDH neurons) showing ACC electrical stimulation increased the frequency and amplitude of pinch induced EPSC. *frequency: $p = 0.04$; amplitude: $p = 0.03$, paired $t$-test. **c** One sample and summarized results (7 SDH neurons) showing ACC electrical stimulation did not affect the frequency and amplitude of puff induced EPSC. **d** One recording sample and summarized results (7 SDH neurons) showing ACC electrical stimulation did not affect the pinch induced EPSC

Meanwhile, we did not find that ACC stimulation directly induce transient $Ca^{2+}$ responses.

Brushing stimuli induced faint or no response in most of the GCaMP6-expressing neurons. However, after ACC stimulation, more than half of the responsive neurons exhibited significantly potentiated responses to brushing stimuli (Fig. 8c) (Supplementary movie 2). In total of 85 identified neurons, 33 (37.7%) showed responses to brushing stimuli. After ACC stimulation, the

$Ca^{2+}$ activity from 63.6% of these responsive cells was potentiated, from 27.3 to 6.1% of the responsive neurons was alleviated or unchanged (Fig. 8e). Finally, the alteration ratios of the calcium responses seemed to have large varieties ranging 30–680% increase in response to pinching ($205 \pm 250$% in average) and 21–745% increase in response to brushing ($272 \pm 143$% in average) (Fig. 8d). In accordance with our in vivo patch recording results, the $Ca^{2+}$ imaging results indicate that electrical

activation of ACC should modulate the responsiveness of SDH neurons, and the direction of the modulation is biased to potentiate neuronal responses to peripheral sensory inputs.

**ACC projection synapses with spinal superficial neurons.** Our functional results indicate that ACC directly potentiate the SDH

neurons, we then feel interested to observe the morphological connections between ACC descending fibers and SDH neurons. An anterograde tracer PHA-L was injected into one side of the ACC (Supplementary Fig. 3a) and PHA-L-labeled fibers and terminals were observed in the SDH. These descending fibers and terminals were observed bilaterally with a contralateral

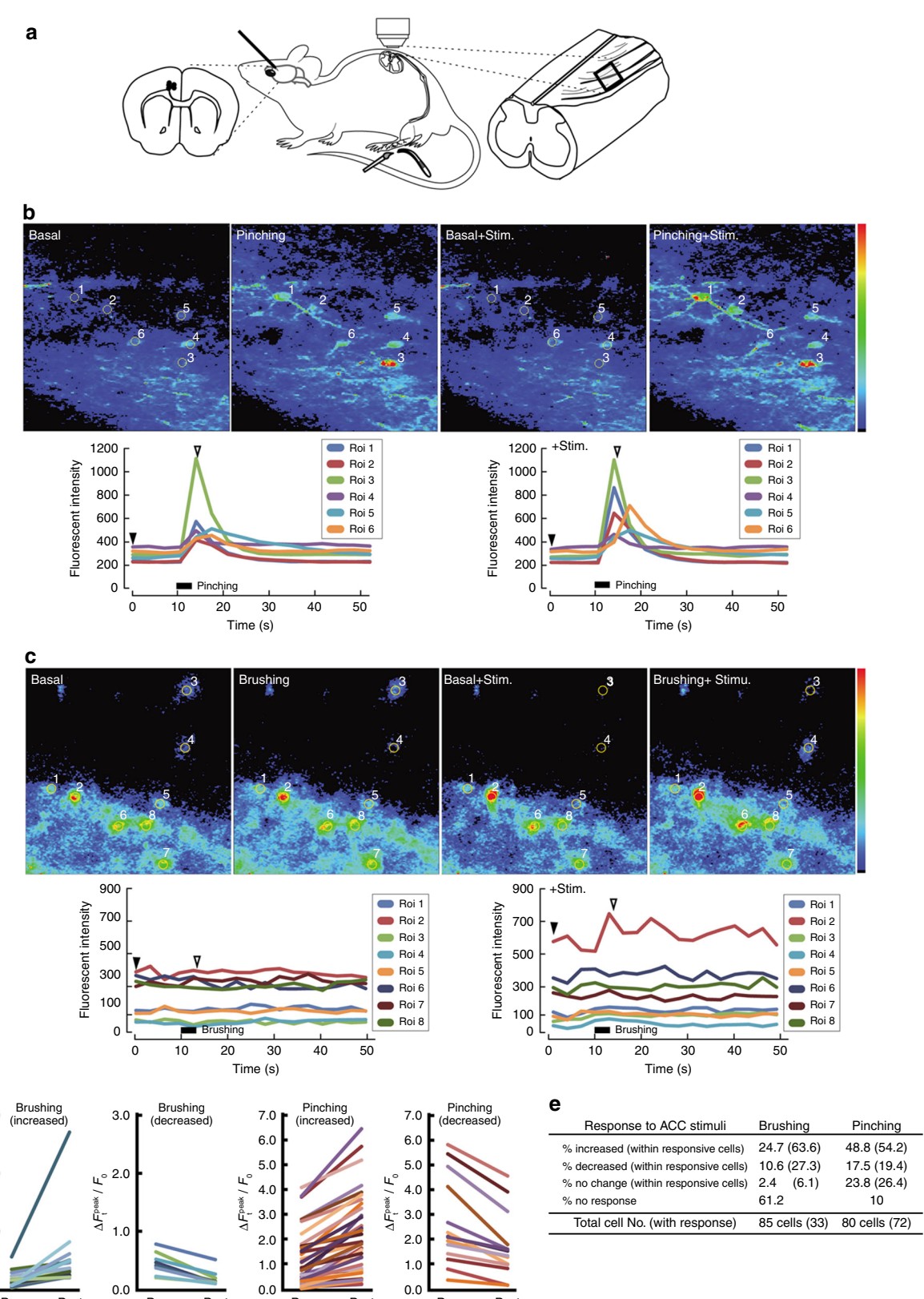

dominance in the spinal superficial laminae, with most of the varicose and punctate fibers and terminals distributed in the laminae I/II and scattered in the laminae III–IV (Supplementary Figs. 3b-i).

These results suggest that ACC-spinal cord projecting fibers should make connections with both projecting neurons and local neurons in spinal laminae I/II. To confirm this, we first labeled lamina-I projecting neurons for fluoro-gold (FG) injection into the lateral parabrachial nucleus (LPB) (Supplementary Fig. 4a), the ascending target for more than 95% lamina-I projecting neurons[34]. Under confocal microscopy, FG retrogradely labeled neurons were mainly located on the contralateral SDH to the injection site, concentrated in lamina I and scattered in laminae III–VI (Supplementary Fig. 4b). Meanwhile, more than half (52.4%) of the PHA-L-labeled fibers were found in close apposition to the FG-labeled neurons in lamina I (Fig. 9a, Supplementary Figs. 4c-h), suggesting that ACC-spinal cord descending fibers should connect with the lamina-I projecting neurons. To further confirm this synaptic connection, we subsequently performed EM works. In this case, ACC-spinal cord projecting fibers and terminals were marked with PHA-L, while LPB-projecting lamina I neurons were marked with HRP. Under electron microscopy, we observed that dense PHA-L-labeled axonal terminals were distributed in lamina I. Within 130 randomly encountered PHA-L-labeled axonal terminals in lamina I, 98 (74.2%) made synaptic connections with HRP-labeled soma or dendrites and only 32 (25.8%) made synaptic connections with immunonegative post-synaptic profiles. Furthermore, within the total 130 synapses, 119 were asymmetric and only 11 were symmetric (Fig. 9c–d). Since asymmetric synapse in spinal cord is normally considered as excitatory, the present results strongly suggest that ACC-spinal cord projecting fibers could directly excite the lamina-I projecting neurons.

Due to the existence of numerous inhibitory interneurons within the SDH, we further investigated the connections between PHA-L-labeled fibers and GAD67 immunoreactive (IR) interneurons within laminae I/II. The GAD67 antibody identifies the 67 isoform of GAD and considered as a marker for GABAergic inhibitory interneurons. Our immunofluorescent staining results showed GAD-IR neurons were concentrated in lamina II and scattered in laminae I, III–IV, in consistence with our previous reports[39]. However, only a few (10.7%) PHA-L-labeled fibers within lamina II made close contact with GAD67-IR neuronal soma (Fig. 9b and Supplementary Figs. 4i-n). Within lamina I, we did not found identified close contact. We then checked the synaptic connections between PHA-L-labeled axon terminals and GAD-IR soma and dendrites in laminae I and II. We found that, in lamina II, 14.9% (10/67) of randomly selected PHA-L-labeled axonal terminals made synapses with GAD-IR soma or dendrites but the left 85% (57/67) of axonal terminals contacted with immunonegative post-synaptic profiles (Fig. 9e–f). In lamina I, only 6 of 115 PHA-L-labeled axonal terminals contacted with GAD-IR profiles but the left contacted with immunonegative profiles. Finally, most of the synapses (170/182) were asymmetric. These results suggest that ACC-spinal cord projecting fibers may

preferentially excite the excitatory but not inhibitory interneurons in SDH.

**Activation of ACC-SDH projection induced pain sensitization.** Our morphological and functional results indicated that ACC potentiated the spinal excitatory transmission, suggesting this top-down potentiation may contribute to chronic pain condition. We thus employed optogenetic technique to determine whether manipulation of the activity of ACC-spinal cord pathway may change the behavioral pain responses. We firstly injected the retrograde labeling tracer canine adenovirus-2 expressing Cre (CAV2-Cre) into the SDH of Ai 32 (Rosa26-stop$^{flox}$-ChR2 (H134R)-EYFP) or Ai35 (Rosa26-stop$^{flox}$-Arch-GFP) mice, in which the CAV2-cre retrogradely infected neurons can express ChR2-EYFP (in Ai32 mice) or Arch-GFP (in Ai35 mice). Four weeks after CAV2-cre injection, we observed EYFP-expressing neurons or GFP-expressing neurons in both sides of the ACC in Ai32 or Ai35 mice (Fig. 10a), indicating the stable infection of CAV2-cre on ACC-spinal cord projecting neurons.

We then implanted optical fiber into the ACC and tested whether activation of the ACC-spinal cord projecting neurons would affect the nociceptive responses (Fig. 10b–c). We found that, during the blue light (490 nm) "on" session, the mechanical withdrawal thresholds were significantly decreased in both left and right hind paws of Ai32 mice with CAV2-cre infection (Ai32 +cre), in comparison with the thresholds in light "off" session (left paw: off, $0.36 \pm 0.08$ g; on, $0.17 \pm 0.08$ g, $p = 0.001$; right paw: off, $0.37 \pm 0.08$ g; on, $0.18 \pm 0.07$ g, $p < 0.01$, $n = 6$ mice). This effect was replicated when more sessions of light excitation was employed. By contrast, blue light failed to affect the withdrawal thresholds in Ai32 mice without CAV2-cre infection (Ai32-cre) (left paw: off, $0.43 \pm 0.06$ g; on, $0.37 \pm 0.08$ g, $p > 0.05$; right paw: off, $0.47 \pm 0.05$ g; on, $0.40 \pm 0.05$ g, $p > 0.05$, $n = 6$ mice) (Fig. 10d).

We then tested whether inhibition of the ACC-spinal cord pathway could affect the nociceptive responses (Fig. 10e–f). We found that continuously yellow light (590 nm) stimulation in the "on" session had no effect on the hind paw withdrawal thresholds in normal Ai35 mice with CAV2-cre infection (Ai35+cre) (left paw: off, $0.41 \pm 0.05$ g; on, $0.39 \pm 0.08$ g, $p > 0.05$; right paw: off, $0.42 \pm 0.07$ g; on, $0.41 \pm 0.08$ g, $p > 0.05$, $n = 5$ mice), neither in normal Ai35 mice without CAV2-cre infection (Ai35-cre) (left paw: off, $0.41 \pm 0.04$ g; on, $0.43 \pm 0.08$ g, $p > 0.05$; right paw: off, $0.38 \pm 0.05$ g; on, $0.39 \pm 0.08$ g, $p > 0.05$, $n = 5$ mice) (Fig. 10g). However, in these mice with neuropathic pain after bilateral common peroneal nerve (CPN) ligation, yellow light stimulation in the ACC-spinal cord neurons caused obvious analgesic effect in Ai35+cre (left paw: off, $0.05 \pm 0.03$ g; on, $0.19 \pm 0.06$ g, $p < 0.05$; right paw: off, $0.09 \pm 0.05$ g; on, $0.28 \pm 0.11$ g, $p < 0.05$, $n = 5$ mice) but not in Ai35-cre mice (left paw: off, $0.04 \pm 0.02$ g; on, $0.07 \pm 0.03$ g, $p > 0.05$; right paw: off, $0.07 \pm 0.04$ g; on, $0.11 \pm 0.05$ g, $p > 0.05$, $n = 5$ mice) (Fig. 10h). In sum, the optogenetic works indicate that activation of ACC-spinal cord pathway facilitated nociceptive responses, while inhibition of ACC-spinal cord

**Fig. 8** In vivo $Ca^{2+}$ imaging of spinal dorsal horn neurons. **a** A scheme of experimental setup. Two-photon imaging of SDH was performed through custom built implanted imaging chamber and electrode was inserted into one side of ACC after a small craniotomy. Pinching and brushing were applied to the contralateral hindpaw. **b**–**c** Representative image showing the $Ca^{2+}$ responses before and after pinching/brushing and ACC stimuli. Raw fluorescent intensity was pseudo-colored and region of interests were numbered. Changes of intensity were plotted under the images and corresponding time points to images were indicated by open arrow head (Pinching/Brushing) and closed arrow head (basal). Roi 1, 2, 6 in **b** and Roi 1, 2, 4 in **c** were identified neurons with potentiated responses to ACC stimulation; Roi 3 or Roi 4, 5 in **b** were neurons with alleviated or no changed responses. Roi 3, 4, 6–8 in **c** have no responses to brushing. **d** Maximal $Ca^{2+}$ response to brushing and pinching in each cells indicated as $\Delta F_t/F_0$ were plotted at pre-ACC and post-ACC stimulation. Data were separately plotted on the basis of the effect of ACC stimulation (increase or decrease). **e** Summarized results of identified cells before and after pinching/brushing and ACC stimuli

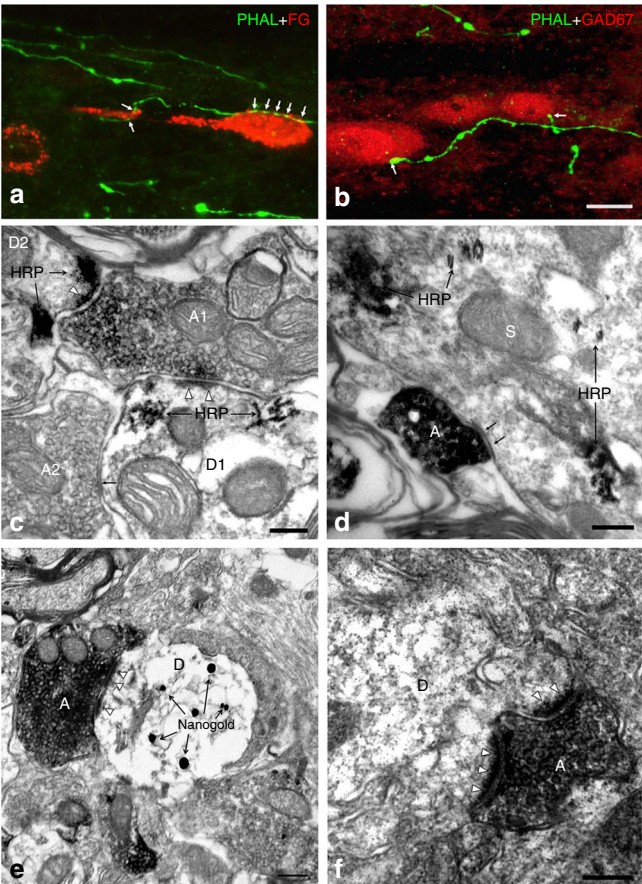

**Fig. 9** ACC-spinal cord projecting fibers and terminals connected with SDH neurons. **a–b** One Alexa-488-labeled PHA-L-IR fiber (for PHA-L injected into the contralateral side of ACC) made close connection with one cy3 re-stained FG-labeled spinal lamina I neuron (for FG injection into the contralateral PBN) (**a**) or two cy3-labeled GAD67-IR neurons in spinal lamina II (**b**). **c** One DAB stained PHA-L containing axon (A1) made asymmetric synapses with two HRP-labeled dendritic spines (D1, 2), noting that the axon terminal mainly contained round and clear vesicles. An immunonegative axon (A2) also made symmetric synapse with D1, noting the axon terminal mainly contained flat and clear vesicles. **d** One PHA-L containing axon made symmetric synapse with one HRP-labeled soma (S). **e–f** One PHA-L containing axon made asymmetric synapses with one nanogold-labeled GAD67-IR (**e**) or immunonegative dendritic spine (**f**). Arrows in **a–b** indicate the close connection sites, in **c–d** indicate the symmetric synapses. Triangles point to the asymmetric synaptic regions. Bars equal to 10 μm in **a–b**, 200 nm in **c–f**

pathway alleviated the nerve injury-induced behavioral sensitization.

## Discussion

In the present study, by using in vivo whole-cell patch-clamp recording and Ca$^{2+}$ imaging works, we directly show that stimulation of ACC neurons enhance the spinal sensory excitatory transmission and neuronal activity, and the facilitatory pathway from ACC-spinal cord is likely not affected by RVM blockade. Anatomic evidence showed that this top-down facilitation may due to the direct excitatory contacts from ACC descending fibers to the SDH projecting neurons and excitatory interneurons. Finally, optogenetic findings provide strong evidence that activation of the ACC-spinal cord pathway contributes to the nerve injury-related behavioral sensitization. This newly discovered cortical-spinal cord descending pathway provides a long range

facilitation of spinal nociceptive transmission and a new mechanism for ACC's modulatory effect in nociceptive responses.

It is well known that spinal nociceptive transmission can be regulated by different supraspinal brain structures. Activation of neurons in the brainstem RVM produces biphasic modulation of spinal nociceptive transmission and behavioral nociceptive reflexes[1–3]. Unlike brainstem, few studies have suggested that cortex may affect spinal nociceptive transmission. According to our previous works, it is found that electrical stimulation of ACC can facilitate behavioral nociceptive TF reflex[23], suggesting a corticospinal top-down modulation. However, these results are mainly based on behavioral observation. It lacks direct morphological and functional evidences to determine if such corticospinal modulation is due to the direct affection on SDH neurons, as well as changing the spinal synaptic transmission. The present study demonstrates that ACC stimulation facilitates glutamate-mediated excitatory transmission in the SDH neurons, directly confirming the facilitatory modulation of spinal nociceptive transmission by stimulating cortex.

It is important to know if ACC produced facilitation is mediated through lower subcortical nuclei, despite the descending projections from the ACC to the spinal cord. Previous studies found that ACC produced descending facilitation of pain behavior that require RVM relay[23]. This is also shown by anatomic evidence that ACC neurons project to the brainstem nuclei, such as periaqueductal gray and RVM nuclei[35]. By contrast, we found that ACC stimulation-induced facilitation of spinal sensory transmission is independent of the RVM. RVM blockade, either before or after ACC stimulation, cannot block or reverse the ACC stimulation-induced facilitation of spinal sEPSC. Selectively optogenetic manipulation of ACC-spinal cord projecting neurons provided a strong evidence for the RVM-independent ACC-spinal cord pathway in pain modulation. Further considering the present Fos staining results, we may explain this as that ACC facilitation affect the activity of laminae I–II neurons of SDH, which is also consistent with our present and previous data that ACC mainly projects to the laminae I–II[24,38], while RVM descending projection more likely affect the deep laminae of the SDH[35,40]. However, drawing a conclusion of the RVM's effect on spinal cord should be very careful, since stimulation of RVM induced both facilitation and inhibition of pain behaviors[5–7] and excitation or inhibition of RVM also induced increased and decreased expression of Fos in SDH[36]. A more safe explanation for the ACC-induced descending facilitation could be there are two facilitatory pathways: ACC-RVM-spinal cord and ACC-spinal cord pathways. They may work as parallel descending modulatory systems and are not depend on each other. We also propose these two parallel descending facilitatory pathways should offer complementary mechanism to "gate" sensory transmission in the spinal level: the ACC-spinal cord pathway offers rapid enhancement since glutamate released from ACC projecting neurons may acutely affect spinal transmission; ACC-RVM-spinal cord pathway allows a prolonged enhancement since serotonin released from RVM is the major mediator for long term facilitatory effects in spinal transmission[1,4,41].

Synaptic mechanisms in ACC involved with potentiated cortical neuron excitability for chronic pain modulation have been extensively studied[24,42–44]. After peripheral inflammation or nerve injuries, increased expression of GluA1 have been confirmed to respond to the potentiated neuronal activity in ACC[24,38,42,44]. In the present study, we found that inhibiting the GluA1 expression in the ACC in turn reduced the frequency of sEPSC in SDH neurons. Considering glutamate is the major transmitter for pyramidal cells in the ACC, it raises the possibility that ACC-spinal cord output neurons release glutamate as a transmitter. We found that the ACC-spinal cord projecting axon

terminals contained round and clear vesicles and mainly made asymmetric synapses with post-synaptic profiles, which is the typical characteristic of glutamatergic axon terminals in the spinal cord. The released glutamate may thus act on different types of glutamate receptors, such as NMDA, recruited AMPA, kainite, and metabotropic glutamate receptors, on the SDH neurons and

amplify the neuronal excitation and synaptic transmission[32,45,46]. In our previous works, we have shown that microinjection of GluA1 antagonist NASPM or cAMP inhibitor Rp-cAMP into the layer V (output layer) of ACC can reverse the behavioral allodynia effect induced by peripheral nerve injury[38]. In the present study, inhibition of the Arch-expressing ACC-spinal cord

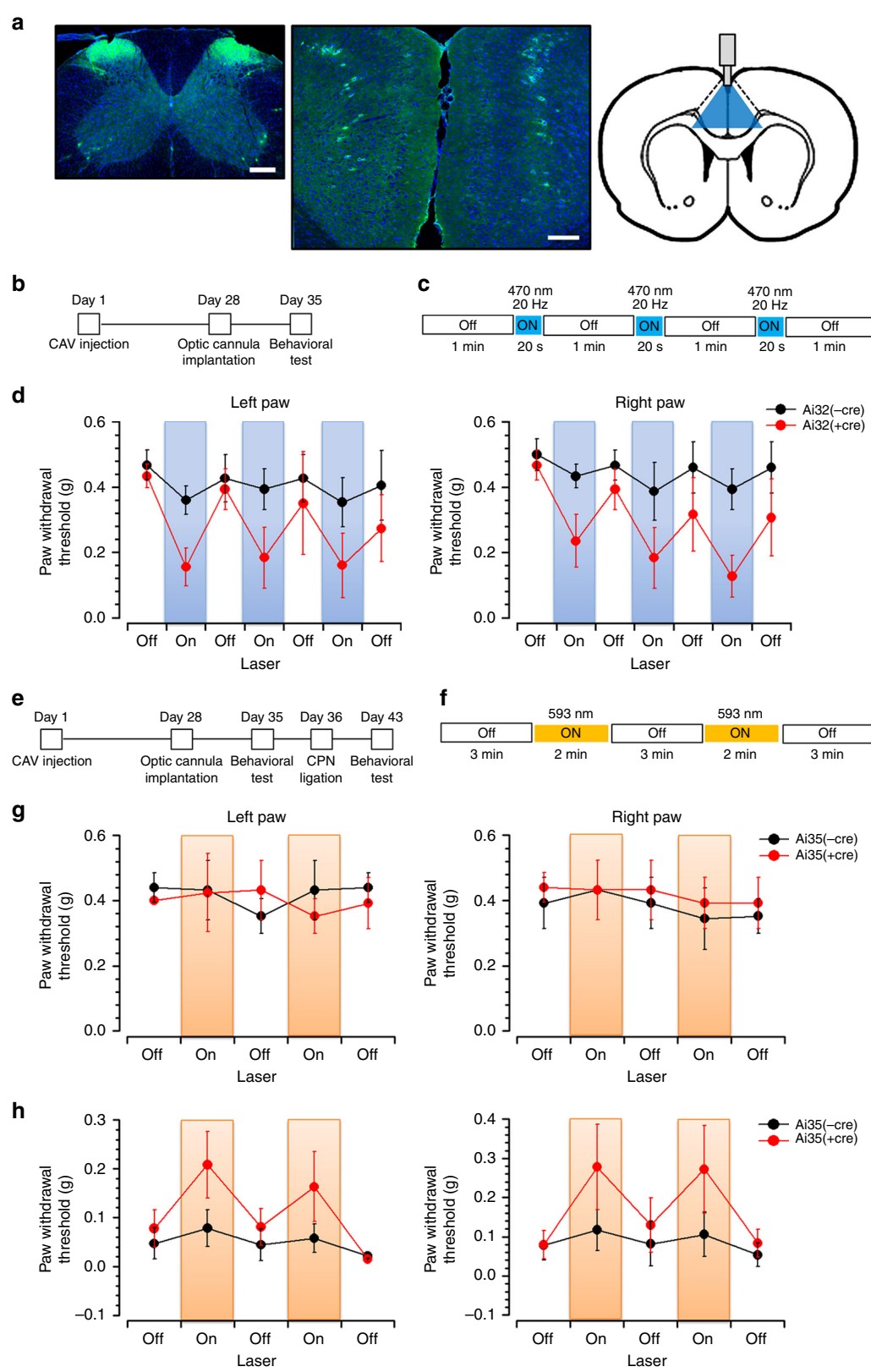

projecting neurons caused the similar analgesic effect. We think this may come from the direct inhibition of the ACC descending fibers and reduced glutamate release to the spinal cord neurons, as shown in the electrophysiological results.

Descending facilitation has been recently indicated to play key roles in central sensitization for nociceptive inputs and maintenance of chronic pain in different animal models[2,8]. However, the circuitry connection and mechanism for descending facilitation that lead to the development of chronic pain conditions remain obscure and far from well understood. In the present study, we found activation of ACC-spinal cord pathway potentiated the synaptic transmission and neuronal $Ca^{2+}$ responses in the SDH and induced pain sensitization in naïve or sham operated animals. Meanwhile, alleviating the ACC-spinal cord output reduced the nerve injury potentiated spinal synaptic transmission and behavioral pain sensitization. Thus, it is possible that ACC-spinal cord pathway function for the enhancement of spinal sensory information in physiological condition. In chronic pain condition, the persistently activated ACC neurons will cause tonic potentiation effect on the spinal sensory transmission and contribute to the maintenance of behavioral hyperalgesia[24,38]. Considering that ACC converges the emotional and sensory information, it is also possible that the top-down pathway contributes to emotional pain, such as phantom pain. We hypothesize that emotional changes may trigger abnormal enhancement of synaptic transmission in the ACC and thus increased the activity of dorsal horn neurons through the top-down pathway. Both the potentiated spontaneous and stimulation-induced activity in spinal cord neurons will in turn cause pseudo "pain" like sensation without noxious peripheral inputs.

In sum, these findings provide insights for designing new treatment methods and protocols, as well as exploring possible novel targets for analgesic drugs. Potentially, one may reduce chronic pain or emotional pain by inhibiting injury triggered potentiation in the cortex, and/or inhibiting corticospinal descending facilitation. Further studies are clearly needed to identify the transmitters and mechanisms for such descending facilitation in pathological pain conditions.

## Methods

**Animals**. Male Sprague-Dawley rats (5–6 weeks) and male mice (8–10 weeks) (C57, Ai32 (Rosa26-stop^flox-ChR2 (H134R)-EYFP) or Ai35 (Rosa26-stop^flox-Arch-GFP)) were used. Animals were randomly housed under a 12-h light-dark cycle (9 a.m. to 9 p.m. light), with food and water freely available, at least one week before carrying out experiments. Experimental procedures involving animals were approved by Animal Care and Use Committee of the Fourth Military Medical University, Xi'an Jiaotong University, Kyushu University, or Kansai University of Health Sciences.

**Nerve injury model**. A model of neuropathic pain in rats was induced by the ligation of the tibial and CPN as described previously[47]. Briefly, after anesthetized with ketamine hydrochloride (45 mg/kg, i.p.) and xylazine (10 mg/kg i.p.), the skin on the lateral surface of the left thigh was incised and a section made directly through the biceps femoris muscle exposing the sciatic nerve and its three terminal branches: the sural, common peroneal, and tibial nerves (Fig. 2). The SNI procedure comprised an axotomy and ligation of the tibial and CPN leaving the sural

nerve intact. The common peroneal and the tibial nerves were tight ligated with 5-0 silk and sectioned distal to the ligation, removing 2–4 mm of the distal nerve stump. Great care was taken to avoid any contact with or stretching of the intact sural nerve. Muscle and skin were then sutured in two layers and cleaned. Sham operation was conducted in the same manner, leaving the nerve without lesion.

A model of neuropathic pain in mice was induced by the ligation of the CPN. Briefly, mice were anesthetized by an intraperitoneal injection of a mixture saline of ketamine (0.16 mg/kg) and xylazine (0.01 mg/kg). The CPN was visible between the anterior and posterior groups of muscles, running almost transversely. Bilateral CPNs were slowly ligated with chromic gut suture 5-0 until contraction of the dorsiflexor of the foot was visible as twitching of the digits. The skin was then sutured and cleaned. The mice were used for behavioral test on post-surgical days 7.

**In vivo patch recording**. The methods used for the in vivo whole-cell patch-clamp recording from SDH neurons were similar to those described previously[48]. Rats were anesthetized with urethane (1.2–1.5 g/kg, i.p.). The spinal cord was exposed at the level from L3 to L5 by a thoraco-lumbar laminectomy, and the rat was placed in a stereotaxic apparatus. Then the dorsal root that enters the spinal cord above the level of recording sites was gently shifted bilaterally, using a small glass retractor, to expose Lissauer's tract so that a recording electrode could be advanced into the SDH from the surface of the spinal cord. The pia-arachnoid membrane was removed using microforceps to make a window large enough to allow the patch electrode to enter the spinal cord. The surface of the spinal cord was irrigated with 95% $O_2$ 5% $CO_2$-equilibrated Krebs solution (10–15 ml/min) (mM: NaCl 117, KCl 3.6, $CaCl_2$ 2.5, $MgCl_2$ 1.2, $NaH_2PO_4$ 1.2, glucose 11, and $NaHCO_3$ 25) through glass pipettes at 36.5 ± 0.5 °C. The patch-electrodes were filled with a patch-pipette solution used to record EPSCs composed of the following (mM): K-gluconate 135, KCl 5, $CaCl_2$ 0.5, $MgCl_2$ 2, EGTA 5, ATP-Mg 5, and HEPES-KOH 5, pH 7.2, and used to record IPSCs composed of the following (in mM): 110 $Cs_2SO_4$, 5 tetra-ethylammonium, 0.5 $CaCl_2$, 2 $MgCl_2$, 5 EGTA, 5 HEPES, and 5 ATP-Mg, pH 7.2. 0.5% biocytin was added into the pipette solution for labeling the recorded SDH neurons. The electrode with a resistance of 8–12 MΩ was advanced at an angle of 30–45 degrees into the SDH through the window in the pia-arachnoid membrane using a micromanipulator (MWS-32S, Narishige, Tokyo, Japan). A giga-ohm seal (resistance of at least 10 GΩ) was then formed with neurons at a depth of 30–150 μm. Cells with resting membrane potential < −50 mV were selected for following experiments and the membrane potentials were held at −70 mV in voltage-clamp mode. Signals were collected using an Axopatch 200B amplifier in conjunction with a Digidata 1440AA/D converter and stored on a personal computer using pCLAMP 10 data acquisition program.

The methods used for electrical stimulation or NASPM injection in the ACC, and lidocaine/CNQX injection in the RVM during in vivo patch-clamp recording on SDH neurons were similar to those described previously. A burr hole was made in the skull and a concentric bipolar stimulating electrode (0.15 mm OD; model MS308/SPC; PlasticsOne, Roanoke, USA) or Hamilton syringe attached to a 30-gauge injector was inserted. The stimulating and injection sites were aimed at contralateral ACC (stereotaxic coordinates: 2.0 mm anterior to the bregma and 0.5 mm lateral, 2.0 mm ventral to the dura) and middle or both sides of the RVM (stereotaxic coordinates: 11.4 mm posterior to the bregma and 10.0 mm ventral to the skull, 0.4 mm lateral to the midline). The electrical stimulation delivered to the ACC was performed with rectangular pulses (pulse duration, 100 μs; intensity, 100 μA; frequency, 100 Hz) that lasted for 1 s. They were repeated every 5 s and the total stimulation periods were 2.5 min.

After getting stable recording for one spinal cord neuron, we apply ACC stimulation and observe the stimulation-induced effect on the spinal neuron. Whether the whole procedure is successfully finished or not, we will not record more neurons in the same animal since ACC stimulation may change the basal activity of the other unpatched spinal neurons. Thus, the numbers of recorded neurons and used animals are the same.

**In vivo two-photon $Ca^{2+}$ imaging experiments**. The gene encoding GCaMP6s (addgene #40753) were subcloned into the pENTR plasmid. The GCaMP6s cassette was transferred into the AAV shuttle vector (pZac2.1, provided by the Vector Core of the University of Pennsylvania) with ESYN, an enhanced human synapsin promoter (addgene #32581), and the woodchuck post-regulatory element (WPRE)

**Fig. 10** Activation of ACC-spinal cord projecting neurons contributed to mechanical pain sensitization. **a** Photos showing the CAV2-cre injection sites in bilateral spinal dorsal horns (left) and the CAV2-cre infected neurons in both sides of the ACC (middle) in Ai32 mice. Diagram showing the optic cannula implantation in the middle side of the ACC. Bars equal to 200 μm. **b** Schematics showing the timeline of experiments for CAV2-cre injection into the spinal cord, optic cannula implantation into the ACC, and behavioral test in Ai32 mice. **c** The procedure for optogenetic modulation during the mechanical pain threshold test. Blue light stimuli (470 nm, 20 Hz) were delivered during the "on" session and repeated for 3 time with 1 min intervals. **d** Blue light stimulation reduced the paw withdrawal threshold in both left and right hind paws of Ai32 (+cre) but not Ai32 (−cre) mice. **e** Schematics showing the timeline of experiments in Ai35 mice. **f** The procedure for optogenetic modulation during the mechanical pain threshold test. Yellow light stimuli (590 nm) were delivered continuously during the "on" session and repeated for 2 time with 3 min interval. **g** Yellow light stimulation had no effect on the paw withdrawal threshold in normal Ai35 (+cre) and Ai35 (−cre) mice. **h** Yellow light stimulation inhibited the allodynia in Ai35 (+cre) but not Ai35 (−cre) mice after CPN ligation

sequence (pZac2.1-ESYN-GCaMP6s-WPRE). AAV vectors were co-transfected with AAV serotype 9 helper plasmid into human embryonic kidney 293 (HEK293) cells and were purified by two rounds cesium chloride density gradient purification steps. The vector was dialyzed against phosphate-buffered saline (PBS) containing 0.001% (v/v) Pluronic-F68 using Amicon Ultra 100 K filter units (Millipore, Darmstadt, Germany). Viral titer was determined by Pico Green fluorometric reagent (Molecular Probes, OR, USA). Viral aliquots were stored at −80 °C until use. Under anesthesia with ketamine (100 mg/kg) and xylazine (10 mg/kg), 500 nl of AAV ($1 \times 10^{13}$ viral genomic copies/ml) was delivered with a glass micro-capillary into the left side of SDH between Th13 and L1 vertebrae (200 μm in depth from the dura)[49].

SDH neurons were affected with AAV9-ESYN-GCaMP6s-WPRE in one side of SDH (L4-5) in adult mice. Three to four weeks after GCaMP6s transduction with AAV, mice were deeply anesthetized by subcutaneous injection of ketamine (100 mg/kg) and xylazine (10 mg/kg), and custom made spinal cord imaging chamber was attached to vertebrae at T12-L1 as previously reported with minor modifications[50]. The skin was incised at Th11-L2. The paravertebral muscle overlying Th12-L1 was removed and tendons are severed from the transverse processes. The laminectomy was performed at Th13, and a custom-made spinal chamber was attached on the laminectomy. Imaging window was covered with Kwik Sil elastomer, and a glass coverslip was placed to seal the window. For ACC stimulation, a craniotomy (1 mm in diameter) centered 1.5 mm anterior to the bregma and 0.3 mm lateral to the midline was made using a dental drill.

Two-photon $Ca^{2+}$ imaging was performed under isoflurane anesthesia by using an Olympus FV1000 with a $25 \times 1.05$ NA water-immersion lens (Olympus) and Spectra Physics Mai-Tai IR laser tuned at 900 nm for two-photon excitation for GCaMP6s. During two-photon imaging, mice were placed on a heating pad. $Ca^{2+}$ imaging was performed on the neurons that located at a depth of 25–70 μm of one side of lumber spinal cord. The electrode for ACC stimulation was inserted through craniotomy to place electrode tip in the ACC area contralateral to the spinal imaging side (stereotaxic coordinates: 1.5 mm anterior to the bregma and 0.3 mm lateral, 1.0 mm ventral to the dura). After 10 min with confirmation of no bleeding from electrode insertion site, peripheral sensory stimuli were applied to the hindpaw using brush and subsequently forceps. An electrical stimulation delivered to the ACC was performed with rectangular pulses and lasted for 300 ms (pulse duration, 100 μs; intensity, 100 μA; frequency, 100 Hz), and after 2 min, peripheral sensory stimulus (3 s) was applied to the left hindpaw again. Obtained image data ($512 \times 512$ pixels, 3.5–5 s/frame) were analyzed with Image J (http://imagej.nih.gov/ij/). The fluorescent signals were quantified by measuring the mean pixel intensities of the cell body of each neurons. The fluorescent change was defined as $\Delta F/F_0 = (F_t - F_0)/F_0$. $F_t$ was the fluorescent intensity at time t, and $F_0$ was the baseline intensity obtained by first value of pre-stimulus. The fluorescence intensity changed less than 20% from initial response is considered no change.

The noxious and innocuous mechanical stimuli were applied to the receptive field of the hindlimb ipsilateral to the $Ca^{2+}$ responses recording neurons by using toothed forceps, brush or air puffs (Pressure system IIe, Toohey Company, Fairfield, NJ, USA), respectively. The brush, air puff, and pinch stimuli were applied at least three times before and after ACC stimulation and the duration for each stimulation lasted for 5 s. To keep a fixed strength noxious stimulation, the toothed forceps were clamped during skin pinching.

**Immunohistochemistry staining**. Retrograde tracer HRP and FG was injected into the PBN to label spinal lamina-I projecting neurons. Anterograde tracer PHA-L was injected into the ACC for tracing ACC-originating fibers and terminals in the spinal cord. The procedure for retrograde/anterograde tracers injection into the brain was according to our previous works[24].

For visualization of the biocytin-labeled SDH neurons, rats were perfused with 0.1 mol/L PBS and 4% paraformaldehyde immediately after in vivo recording. After fixation, L4-L5 spinal cords were removed, cryoprotected, and serially cut into transverse slices with 30 μm thickness. The sections were then rinsed in PBS with 0.3% Triton X-100 and 1% normal bovine serum, incubated with avidin–biotin complex (ABC) solution (Elite ABC kit; Vector Laboratories, Burlingame, CA, USA) for 2 h. After thoroughly rinses, the sections were reacted with 0.05% diaminobenzidine (DAB) and 0.003% $H_2O_2$ in PBS to visualize biocytin labeled neurons. After washing, staining was observed under microscopy (Olympus BX61, Tokyo, Japan).

For the EM staining and observation, rats were deeply anesthetized and then transcardially perfused with 0.9% saline, followed by 0.1 M PB containing 4% paraformaldehyde, 0.1% glutaraldehyde, and 15% picric acid. Sections from the spinal cord were generated using a vibratome (Microslicer DTM-1000, DSK, Kyoto, Japan) at a 50-μm thickness. The tetramethylbenzidine-sodium tungstate (TMB-ST) method was used to detect HRP[51]. The HRP reaction products were intensified using a DAB/cobalt/$H_2O_2$ solution[52]. The DAB reaction utilized for the demonstration of HRP was performed before the immuno-EM procedures. Details of the immuno-EM procedures for PHA-L and GAD67 were described in our previous reports[53]. Briefly, the sections were incubated in 20% Tris-buffered saline (TBS)-NDS for 30 min to block non-specific immunoreactivity. The sections were incubated for 24 h at 4 °C in primary antibodies to PHA-L and GAD67, respectively. Then, the sections were incubated overnight in biotinylated donkey anti-rabbit IgG for PHA-L or goat anti-mouse IgG conjugated to 1.4-nm gold

particles for GAD67, respectively, at 48 °C. Subsequently, all sections were processed according to the following steps: (1) post-fixation with glutaraldehyde, (2) silver enhancement using the HQ Silver Kit (Nanoprobes, Stony Brook, NY, USA), (3) incubation with the ABC kit (Vector), (4) reaction with DAB tetrahydrochloride and H2O2, (5) osmification, and (6) counterstaining with uranyl acetate. Ultrathin sections at a 70-nm thickness were prepared from the superficial laminae of the SDH, mounted on single-slot grids, and examined under an electron microscope (JEM1440, Tokyo, Japan).

Under the electron microscopy, the HRP-labeled soma and dendritic spine, and shaft were detected based on the presence of highly electron-dense clumps of crystalline material and occasionally the presence of amorphous punctual structures in the cytoplasm and dendrites. The PHA-L-labeled axonal terminals, typically filled with round and clear synaptic vesicles, were ultrastructurally identified by the presence of homogeneously distributed fine, granular, electron-dense reaction products. GAD-IR soma and dendrite were marked with variable number and diameter of gold-silver particles that were mostly distributed within the cytoplasm and scattered within the cellular and organelle membranes.

**Optogenetics**. Retrograde tracer canine adenovirus-2 expressing Cre (CAV2-Cre) were purchased from Taiting Biological Co., Ltd (Shanghai, China). CAV2-cre ($3 \times 10^{12}$, 300 nl per site) were microinjected into the bilateral SDH (C4-5) in adult Ai32 or Ai35 mice. Four to five weeks after injection, mice were anaesthetized and perfused with 0.1 mol/L PBS (pH 7.2–7.4) via the ascending aorta followed by 4% paraformaldehyde in 0.1 M PB (pH 7.4). The spinal cord and brain were then removed, and cryoprotected in 0.1 M PB containing 30% sucrose overnight at 4 °C. Transverse sections (30 μm thickness) of spinal cord and brain samples containing ACC were cut on a freezing microtome and collected serially. Then the sections were used for GFP immunostaining. In brief, sections were sequentially incubated with goat antisera against GFP IgG (1:500, ab104139, Abcam) and Alexa-488 conjugated donkey anti-goat IgG (1:1000, ab150133, Abcam). Sections were then rinsed in PBS, mounted onto glass slides, air dried, cover-slipped with a fluoroshield mounting medium with DAPI (ab104139, Abcam). The signals were visualized under confocal microscope (FV-1000; Olympus, Tokyo, Japan) under appropriate filter for Alexa-488 (excitation 495 nm; emission 519 nm) and DAPI (excitation 360 nm; emission 460 nm).

Four to five weeks after CAV2-cre injection, mice were anesthetized with an intraperitoneal injection of ketaminexylazine (0.1 mg/g body weight ketamine, 0.01 mg/g body weight xylazine) and the head was fixed in a stereotaxic apparatus. A small craniotomy was performed and a hole were drilled. The optic cannula (MFC_200/230-0.39_ 2mm_ZF1.25_FLT, Doric Lenses., Quebec, Canada) was implanted in the middle line of ACC (0.98 mm anterior to Bregma and 1.0 mm deep from skull surface). The optic cannula was then fixed with dental cement.

One week after optic cannula implantation, mice paw withdrawal thresholds were tested with von Frey filaments applied to their left and right hind paws. The mice were placed in Lucite cubicles over a wire mesh and acclimated for 10 min before testing. A series of filaments (0.008, 0.02, 0.04, 0.16, 0.4, 0.6, 1, 1.4, 2 g) with various bending forces (according to 0.078, 0.196, 0.392, 1.568, 3.92, 5.88, 9.8, 13.72, 19.6 mN) were applied to the plantar surface of the hindpaw until the mice withdrew from the stimulus. The lowest force at which a withdrawal response was obtained was considered as the paw withdrawal threshold. In test for Ai32 mice, there were 4 light "off" and 3 light "on" sessions (off-on-off-on-off-on-off) and blue light (470 nm) was given at 20 Hz (40 ms pulse) in an intensity of 15 mW/mm$^2$ at optic fiber tip throughout the "on" session. In test for Ai35 mice, there were 3 "off" and 2 "on" sessions (off-on-off-on-off) and yellow light (590 nm) was given continuously in an intensity of 10 mW/mm$^2$ in the "on" session[54–56]. Paw withdrawal threshold was repeatedly measured in the "off" and "on" sessions and the averages of the 3–4 "off" values or 2–3 "on" values were calculated as their threshold.

**Statistical analyses**. All experiments were carried out as blind to genotype and the conditions of the experiments, unless indicated in naïve animals. Data were collected and processed randomly, and no data points were excluded. No statistical methods were used to predetermine sample sizes, but our sample sizes were similar to those reported in previous publications. Statistical comparisons were made using the unpaired, paired t-test, and one way repeated measures ANOVA (Student–Newman–Keuls test was used for post hoc comparison). The normal distribution and the variation within each group of data was verified by using Sigmaplot 11.0 software before applying statistical comparison. Analyzed numbers (n) for each set of experiments are indicated in the corresponding figure legends or main text sections. The examples shown in each figure are representative and were reproducible at least three times for each set of experiments. All data were presented as the mean ± S.E.M. In all cases, $p < 0.05$ was considered statistically significant.

**Data availability**. The data that support the findings of this study are available from the corresponding author upon reasonable request.

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

## Acknowledgements

We would like to thank Dr. Si-bo Zhou and Dr. Yu-Peng Feng for technic help in doing imaging and in vivo whole-cell patch works, separately. This work was supported by the Canadian Institute for Health Research (CIHR) Michael Smith Chair in Neurosciences and Mental Health, Canada Research Chair, CIHR Operating Grant MOP-124807, Natural Sciences and Engineering Research Council of Canada Discovery Grant RGPIN 402555, Azrieli Neurodevelopmental Research Program and Brain Canada (M.Z.), The Major International Joint Research Project from NSFC (31010103909) (Y.Q.L. and M.Z.), NSFC (31371126, 81671095 to T.C., and 91732105 to M.Z.), The postdoctoral fellowships from CIHR-FXRFC (K.K. and T.C.), Grants-in-Aid for Scientific Research (25117013) by the Ministry of Education, Culture, Sports, Science, and Technology of Japan (H.T. and K.I.), the program Progress 100 by the Kyushu University, and Health Labour Sciences Research Grant by the Ministry of Health Labour and Welfare (M.T.).

## Author contributions

T.C., Y.Q.L., T.N., and M.Z. designed the experiments. T.C., J.W., Z.H.L., and Y.C.L. did the immunohistochemical staining and EM works. T.C., W.T., Q.Y.C., R.H.L., K.K., T.M., and Y.K.S. performed the in vivo electrophysiological experiments. T.C., H.T.S., and M.T. did the Ca$^{2+}$ imaging works. Q.S. and T.C. did the optogenetic works. T.C. and M.Z. wrote the final manuscript. K.I., Y.Q.L., and T.N. helped to revise the manuscript. All authors discussed the manuscript.

## Additional information

**Competing interests:** The authors declare no competing interests.

