## [Peer Review File · Nature Communications]

Reviewers' comments:

Reviewer #1 (Remarks to the Author):

This article aimed to reveal the direct descending facilitation of spinal synaptic transmission from the anterior cingulate cortex (ACC). By *in vivo* recording, they found sEPSCs and peripheral stimuli-evoked eEPSCs were potentiated following high frequency ACC stimulations. Other studies, including *in vivo* two-photon Ca^{++} imaging, anterograde labeling and EM analyses suggest the existence of a direct ACC-spinal modulation pathway. The findings are potentially interesting, but a number of issues need to be addressed.

1. Earlier studies from these investigators suggested that facilitatory modulation from ACC is relayed via RVM. For this study, the authors emphasized a direct modulation from ACC to superficial spinal neurons. This was concluded by a lack of effect following RVM injection with CNQX. However, the experimental design was flawed. CNQX injection was performed following ACC stimulation, rather than prior to ACC injection. It is quite possible that ACC stimulations may provide glutamatergic excitatory inputs into RVM and sensitize the on-cells in RVM. Once the on-cells were sensitized, CNQX-sensitive glutamate receptors in these neurons might become dispensable. For example, the on-cells might subsequently tonically release serotonin onto the dorsal horn. The authors need to re-do the experiment by injecting CNQX prior to ACC stimulations. The authors should also perform lidocaine injection in RVM post-ACC stimulations to determine if sensitized on-cells in RVMs contributed to increased sEPSCs or eEPSCs in the dorsal horn (CNQX will not be able to block 5-HT release from sensitized on-cells).

2. Will lidocaine injection in RVM block potentiated spinal sEPSC under SNI conditions?

3. The EM work provides the strongest support for direct ACC-spinal inputs. Nonetheless, it would be nice to analyze eEPSCs in dorsal horn neurons during the period of high frequency ACC stimulations to determine if dorsal horn neurons receive monosynaptic inputs from descending ACC neurons.

4. The retrograde labeling and EM studies indicates that descending ACC fibers preferentially form synapses with ascending spinal projection neurons. However, according to Figure 6, more than 50% of dorsal horn neurons may be potentiated. (The authors should clarify the percentages of lamina I/II neurons displaying sensitization following ACC stimulations shown in Figure 1). How could this data be reconciled with the fact that less than 10% of lamina I neurons belong to ascending projection neurons?

5. Figure 7c: D1 was mislabeled as D2.

Reviewer #3 (Remarks to the Author):

This study has explored the potential existence of an RVM-independent descending facilitatory pathway that can enhance sEPSC in the spinal dorsal horn. Key findings are that ACC stimulation potentiated sEPSC's, but did not affect sIPSCs, in naive rats; that inactivation of the RVM with CNQX did not alter descending facilitation from the ACC in naive rats; that this pathway is occluded if descending facilitation is already engaged in nerve injured rats; that inhibition of ACC AMPA receptors with an antagonist blocked increased sEPSCs in SNI rats; that ACC stimulation increased air puff or pinch inputs to SDH as well as potentiation of Ca^{++} responses and finally that ACC projecting fibers made excitatory synapses with lamina I cells backlabeled to project to parabrachial nucleus.

The findings are original and of high significance. Revealing an RVM independent mechanism of potentiation of SDH neurons is important. The authors have combined a number of approaches to test their hypothesis and provide overall support through electrophysiology and anatomy. The suggestion of a "fast" glutamatergic and "slow" serotonergic descending system is novel and intriguing.

There are a number of questions that arise that should be addressed however.

First, the issue of anesthesia in the neuronal responses is not trivial. It is well known that urethane can affect neuronal responses and much of the data are expressed as percent change from baseline. It is possible that a lot of what is being reported may be due to differences in depth of anesthesia and the consequence of anesthesia on nociceptive transmission. This caution in the results must be stated clearly as a limitation of the study. How do the authors control for depth of anesthesia?

Second, the spinal recordings necessarily involve significant injury that may have already induced a state of sensitization as the baseline reading. Inactivation of the ACC with NAPSM decreased the response frequency in SDH of rats with SNI but no data is presented in control rats and it should be.

Third, the duration of the ACC stimulation is not clearly explained. The methods state that rectangular pulses of 100 uS are applied for 300 ms. The figures all show ACC stimulation for 2 minutes. This is confusing. The termination of the ACC stimulation does not result an immediate decrease in the frequency or amplitude response in SDH. The authors do not discuss the duration of the effect that persists for many minutes. How does this occur?

Fourth, the study suffers from a lack of behavioral evaluation. The ACC stimulation is not shown to enhance behavioral responses to stimuli applied to the paws in naive rats and inactivation of the ACC with AMPA antagonist is not shown to produce analgesic responses in SNI rats. The authors cite previous studies in mice but these are inadequate to match the conditions used in this report and this is an important weakness of the study.

Fifth, one of the most important conclusions of the study is that the pathway is independent of the RVM. The evidence for this is delegated to a supplemental figure and is not adequate to support this claim. The authors do not confirm that the RVM is actually blocked by the CNQX and do not perform the experiment in SNI rats. Is there still facilitation from the ACC if the RVM is blocked, a procedure known to alleviate pain?

Sixth, the ACC stimulation does not directly increase Ca^{++} signals in the absence of a noxious input and this is not consistent with the electrophysiological responses. The authors do not comment on this and should.

Seventh, the report mixes response frequency and/or amplitude in terms of number or percent responses, even in the same figure. This is confusing and makes reading difficult. It is not really clear how consistent the facilitation is and what the actual magnitude of the responses are. The figure legends also do not make clear what the N refers to, either numbers of recorded cells or numbers of animals? How many animals were sampled for data from the cell recordings?

Last, the anatomical data are very important and interesting. The ACC projections are demonstrated to contact excitatory ascending neurons, but not inhibitory interneurons. While this is really good, it is not explained how manipulations in the ACC can inhibit von Frey responses after SNI as previously reported by this group. Is this response dependent upon a different (e.g., RVM dependent) pathway? This should be clearly explained in the discussion.

In summary, this is a very strong and important study, but with limitations that should be considered in overall interpretation and conclusions.

Point-to-point responses to Reviewers' comments

Reviewer #1 (Remarks to the Author):

This article aimed to reveal the direct descending facilitation of spinal synaptic transmission from the anterior cingulate cortex (ACC). By in vivo recording, they found sEPSCs and peripheral stimuli-evoked eEPSCs were potentiated following high frequency ACC stimulations. Other studies, including in vivo two-photon Ca^{++} imaging, anterograde labeling and EM analyses suggest the existence of a direct ACC-spinal modulation pathway. The findings are potentially interesting, but a number of issues need to be addressed.

1. Earlier studies from these investigators suggested that facilitatory modulation from ACC is relayed via RVM. For this study, the authors emphasized a direct modulation from ACC to superficial spinal neurons. This was concluded by a lack of effect following RVM injection with CNQX. However, the experimental design was flawed. CNQX injection was performed following ACC stimulation, rather than prior to ACC injection. It is quite possible that ACC stimulations may provide glutamatergic excitatory inputs into RVM and sensitize the on-cells in RVM. Once the on-cells were sensitized, CNQX-sensitive glutamate receptors in these neurons might become dispensable. For example, the on-cells might subsequently tonically release serotonin onto the dorsal horn. The authors need to re-do the experiment by injecting CNQX prior to ACC stimulations. The authors should also perform lidocaine injection in RVM post-ACC stimulations to determine if sensitized on-cells in RVMs contributed to increased sEPSCs or eEPSCs in the dorsal horn (CNQX will not be able to block 5-HT release from sensitized on-cells).

Answer: Thanks for useful suggestion. We have performed additional experiments with lidocaine or CNQX injection as suggested. We firstly injected lidocaine or CNQX into the RVM before ACC stimulation. We found that RVM blockade alone caused either increase or decrease of the sEPSCs; consistent with the fact that RVM exerts biphasic modulation on spinal sensory transmission. After lidocaine/CNQX injection, we then delivered ACC stimulation. ACC stimulation still caused significant potentiation of the sEPSCs; suggesting that ACC induced facilitation is independent of RVM blockade. Secondly, as suggested by this reviewer, we also injected lidocaine after ACC stimulation and found that the potentiation was not affected by lidocaine injection in the RVM. These results further confirm our previous data with CNQX injection.

To extend this finding to spinal circuit, we performed new experiments using Fos staining method. Consistent with electrophysiological results, we found that ACC stimulation activated Fos activity in spinal dorsal horn neurons. Furthermore, lidocaine blockade in the RVM failed to affect the Fos expression triggered by ACC stimulation. These results strongly indicate that ACC stimulation is capable to affect spinal sensory transmission without brainstem relay. These findings, however, do not rule out possible RVM relay in behavioral

responses in whole animals as we previously reported (Calejasan et al., 2000). It is possible that ACC-spinal, and ACC-RVM-spinal pathways may work together in behavioral conditions.

2. Will lidocaine injection in RVM block potentiated spinal sEPSC under SNI conditions?

Answer: As suggested, our new experiments found that lidocaine injection into RVM has no effect on ACC potentiated spinal sEPSC in both sham and SNI groups of rats.

3. The EM work provides the strongest support for direct ACC-spinal inputs. Nonetheless, it would be nice to analyze eEPSCs in dorsal horn neurons during the period of high frequency ACC stimulations to determine if dorsal horn neurons receive monosynaptic inputs from descending ACC neurons.

Answer: Thanks for this interesting suggestion. We have tried to find direct mono-synaptic responses between ACC and spinal cord neurons during stimulation. However, we did not observe the evoked EPSC on recorded spinal neurons according to ACC stimulation. We think it is possible that the projecting fibers in spinal cord is not dense enough for us to detect a direct EPSC. It is possible that released glutamate from ACC projects may act on the metabotropic glutamate receptors (mGluRs). Future experiments are clearly needed to investigate this.

4. The retrograde labeling and EM studies indicates that descending ACC fibers preferentially form synapses with ascending spinal projection neurons. However, according to Figure 6, more than 50% of dorsal horn neurons may be potentiated. (The authors should clarify the percentages of lamina I/II neurons displaying sensitization following ACC stimulations shown in Figure 1). How could this data be reconciled with the fact that less than 10% of lamina I neurons belong to ascending projection neurons?

Answer: From the EM works, we found the ACC-spinal cord projecting fibers/terminals form synaptic connections with both lamina I neurons, and GAD-immunonegative neurons in lamina II. It is possible that ACC-spinal cord projecting fibers may modulate the lamina I projecting neurons as well as lamina II excitatory interneurons. We think this reviewer for excellent observation, and have discussed this in the revised paper. We think that these results match well with the observation shown in Fig. 9 (the Ca²⁺ imaging figure).

5. Figure 7c: D1 was mislabeled as D2.

Answer: Thanks a lot, we have changed that in the new version of Fig. 10c.

Reviewer #3 (Remarks to the Author):

This study has explored the potential existence of an RVM-independent descending facilitatory pathway that can enhance sEPSC in the spinal dorsal horn. Key findings are that ACC stimulation potentiated sEPSC's, but did not affect sIPSCs, in naive rats; that inactivation of the RVM with CNQX did not alter descending facilitation from the ACC in naive rats; that this pathway is occluded if descending facilitation is already engaged in nerve injured rats; that inhibition of ACC AMPA receptors with an antagonist blocked increased sEPSCs in SNI rats; that ACC stimulation increased air puff or pinch inputs to SDH as well as potentiation of Ca⁺⁺ responses and finally that ACC projecting fibers made excitatory synapses with lamina I cells backlabeled to project to parabrachial nucleus.

The findings are original and of high significance. Revealing an RVM independent mechanism of potentiation of SDH neurons is important. The authors have combined a number of approaches to test their hypothesis and provide overall support through electrophysiology and anatomy. The suggestion of a "fast" glutamatergic and "slow" serotonergic descending system is novel and intriguing.

There are a number of questions that arise that should be addressed however.

1. First, the issue of anesthesia in the neuronal responses is not trivial. It is well known that urethane can affect neuronal responses and much of the data are expressed as percent change from baseline. It is possible that a lot of what is being reported may be due to differences in depth of anesthesia and the consequence of anesthesia on nociceptive transmission. This caution in the results must be stated clearly as a limitation of the study. How do the authors control for depth of anesthesia?

Answer: For doing *in vivo* recording on spinal cord, urethane is commonly used (J Physiol. 1999 Dec 1;521 Pt 2:529-35; J Physiol. 2004 Mar 1;555(Pt 2):515-26; Pain. 2011 Jan;152(1):95-105.). It is thus unlikely that ACC stimulation produced effects are simply due to altered anesthesia state, since some manipulation caused opposite changes in spinal neuron activity. We agree with this reviewer that we cannot completely rule out the possible impact of anesthesia level. In some experiments, we also tried some animals with isoflurane anesthesia. We found that ACC stimulation also induced the enhancement of the spinal sEPSC. We thus think that our anesthetic method should not affect the experimental conclusion.

2. Second, the spinal recordings necessarily involve significant injury that may have already induced a state of sensitization as the baseline reading. Inactivation of the ACC with NAPSMM decreased the response frequency in SDH of rats with SNI but no data is presented in control rats and it should be.

Answer: We agreed with the reviewer, and the data is shown in supplementary Fig. 2.

3. Third, the duration of the ACC stimulation is not clearly explained. The methods state that rectangular pulses of 100 uS are applied for 300 ms. The figures all show ACC stimulation for 2 minutes. This is confusing. The termination of the ACC stimulation does not result an immediate decrease in the frequency or amplitude response in SDH. The authors do not discuss the duration of the effect that persists for many minutes. How does this occur?

Answer: Thanks for bring this to our attention. We now added additional information for the stimulation in the revised method. ACC stimulation could induce potentiation of the frequency and amplitude of the spinal sEPSC that lasted for more than 10 min. We propose ACC stimulation may cause a LTP effect on the spinal sEPSC. However, by using *in vivo* whole cell patch method, it is very difficult to record responses for longer period of time. We plan to test the duration effect by using spinal field recording in the future works. We will add this into the revised discussion.

4. Fourth, the study suffers from a lack of behavioral evaluation. The ACC stimulation is not shown to enhance behavioral responses to stimuli applied to the paws in naive rats and inactivation of the ACC with AMPA antagonist is not shown to produce analgesic responses in SNI rats. The authors cite previous studies in mice but these are inadequate to match the conditions used in this report and this is an important weakness of the study.

Answer: Thanks for excellent suggestion. We have decided to perform additional behavioral experiments using the optogenetic method. We first injected retrograde tracer canine adenovirus-2 expressing Cre (CAV-Cre) into the spinal dorsal horn of Ai32 (Rosa26-stop^{flox}-ChR2(H134R)-EYFP) or Ai35 (Rosa26-stopflox-Arch (H134R)-GFP) mice to label the ACC-spinal cord projecting neurons. If labeled successfully, the ACC neurons will express ChR2-EYFP (in Ai32 mice) or Arch-GFP (in Ai35 mice). Four weeks after CAV2-cre injection, we observed EYFP-expressing neurons or GFP-expressing neurons in both sides of the ACC in Ai32 or Ai35 mice, indicating the stable infection of CAV2-cre on ACC-spinal cord projecting neurons. By implanting optical fiber into the ACC, we observed *in vivo* blue light stimulation (470 nm) decreased mechanical threshold, while yellow light stimulation (590 nm) increased pain threshold (new Fig. 11). The new added optogenetic works strongly suggest that activation of ACC-spinal cord pathway directly facilitate nociceptive responses, while inhibition of ACC-spinal cord pathway alleviate the nerve injury-induced behavioral sensitization.

5. Fifth, one of the most important conclusions of the study is that the pathway is independent of the RVM. The evidence for this is delegated to a supplemental figure and is not adequate to support this claim. The authors do not confirm that the RVM is actually blocked by the CNQX and do not perform the experiment in SNI rats. Is there still facilitation from the ACC if the RVM is blocked, a procedure known to alleviate pain?

Answer: As suggested, we performed new experiments and applied lidocaine or CNQX blockade of RVM before or after ACC stimulation in sham operated rats. After lidocaine or

CNQX blockade, ACC stimulation could still cause potentiation effect of the spinal sEPSC. Furthermore, ACC stimulation induced potentiation of spinal sEPSC was not reversed by lidocaine or CNQX blockade in the RVM. As suggested by this reviewer and Reviewer 1, we have also repeated the RVM blockade work in SNI rats. Lidocaine injection in the RVM failed to blocked potentiated spinal sEPSC.

In addition, we have performed new experiments to check the Fos expression in the spinal dorsal horn. We found that ACC stimulation increased the Fos expression in the laminae I-II and RVM blockade did not block the increased expression. These results further support our claim that ACC-spinal pathways may modulate spinal sensory transmission independent of RVM activity.

6. Sixth, the ACC stimulation does not directly increase Ca^{++} signals in the absence of a noxious input and this is not consistent with the electrophysiological responses. The authors do not comment on this and should.

Answer: Thanks for excellent suggestion. We have now added new discussion of these. One possible explanation is that ACC stimulation alone is not sufficient to trigger postsynaptic Ca signaling to be detected, while it does enhance sEPSCs. With peripheral sensory stimuli, ACC stimulation now enhances Ca signaling responses to peripheral stimulation. This is consistent with the fact that ACC stimulation did not cause EPSCs in spinal dorsal horn neurons. It is possible that released glutamate from ACC projects may act on the metabotropic glutamate receptors. We will add these new discussion into the revised paper. Future experiments are clearly needed to investigate this.

7. Seventh, the report mixes response frequency and/or amplitude in terms of number or percent responses, even in the same figure. This is confusing and makes reading difficult. It is not really clear how consistent the facilitation is and what the actual magnitude of the responses are. The figure legends also do not make clear what the N refers to, either numbers of recorded cells or numbers of animals? How many animals were sampled for data from the cell recordings?

Answer: Thanks for suggestion. In the revised version, we fixed the description problem and made it more clear. We have also provided the explanation of the numbers of cells and animals. In this type of in vivo experiments, it is rather technically difficult. After we obtained get stable recording for one spinal cord neuron, we apply ACC stimulation and observe the stimulation induced effect on the spinal neuron. We only record one spinal cord neuron from each animal. Thus, the numbers of recorded neurons and used animals are the same. We have added more description for the experimental procedure in the Method section in the new version of manuscript.

8. Last, the anatomical data are very important and interesting. The ACC projections are

demonstrated to contact excitatory ascending neurons, but not inhibitory interneurons. While this is really good, it is not explained how manipulations in the ACC can inhibit von Frey responses after SNI as previously reported by this group. Is this response dependent upon a different (e.g., RVM dependent) pathway? This should be clearly explained in the discussion.

Answer: In our previous works (Chen et al., Mol Brain, 2014,7:76), we have shown that inhibition of the activity of ACC layer V projecting neurons by microinjection of GluA1 antagonist NASPM or cAMP inhibitor rp-cAMP can reverse the mechanical allodynia effect induced by nerve injury. We think direct inhibition of the activity of ACC-spinal cord neurons will sequentially reduce the glutamate release from the projecting fibers to the spinal cord neurons, as shown in the electrophysiological and optogenetic results, which will finally alleviate pain sensitization. We have thus modified the Discussion part in the new version of manuscript.

REVIEWERS' COMMENTS:

Reviewer #1 (Remarks to the Author):

The authors have carefully carefully addressed my concerns. The study reveals a direct descending facilitation from ACC to spinal neurons, adding a possible spinal-ACC-spinal amplification mechanism for pain information processing. The finding is therefore significant

Reviewer #4 (Remarks to the Author):

In this manuscript, Chen et al. reported a study investigating effects of stimulating the anterior cingulate cortex (ACC) on excitatory synaptic currents in spinal dorsal horn neurons and on behavioral pain responses. It was found that activating ACC by electrical or optogenetic stimulation enhanced excitatory synaptic transmission in projecting neurons and interneurons of dorsal horn in a manner independent of the rostral ventromedial medulla (RVM). Anatomical and optogenetic evidence is provided supporting a direct projection from ACC to the dorsal horn neurons and related synapses. The pain-facilitating effect of ACC activation was shown both in control conditions and under spared nerve injury-induced pain sensitization. Data of Ca^{++} imaging in dorsal horn neurons also support their activation by ACC stimulation.

While descending pain modulation from RVM has been well characterized, a direct pain modulation from cortical areas is rare. In this manuscript, the authors have provided multidisciplinary evidence to demonstrate a direct projection from ACC to neurons in the spinal dorsal horn with presumed excitatory synapses. In the revision, additional data have been added to further support the notion of the ACC effect independent of RVM. Particularly, the new optogenetic data provide much more direct and convincing evidence for the direct pain facilitation by ACC. This represents a significant advance on ACC modulation of pain, and indicates that ACC may have more direct and diverse roles in modulation of sensory pain transmission as well as in modulation of pain-related affect and emotion.

There are some minor issues that need to be addressed to further improve the manuscript.

1. The occlusion of the pain-facilitating effect of ACC stimulation in sensitized pain condition does not necessarily mean the involvement or contribution of ACC (the optogenetic inhibition of the ACC projection does). This is because under pain sensitization, the spinal synaptic transmission studied may have been enhanced enough so that further enhancement by any sites including ACC cannot be observed (ceiling effect). The authors should tune-down the significance of the occlusion experiments (e.g., lines 133-134).

2. Reference should be provided to support the statement "RVM descending projection mainly affect the deep laminae of the SDH" (line 365).

Point-to-point answers:

REVIEWERS' COMMENTS:

Reviewer #1 (Remarks to the Author):

The authors have carefully addressed my concerns. The study reveals a direct descending facilitation from ACC to spinal neurons, adding a possible spinal-ACC-spinal amplification mechanism for pain information processing. The finding is therefore significant.

Answer: Thanks for the support.

Reviewer #4 (Remarks to the Author):

In this manuscript, Chen et al. reported a study investigating effects of stimulating the anterior cingulate cortex (ACC) on excitatory synaptic currents in spinal dorsal horn neurons and on behavioral pain responses. It was found that activating ACC by electrical or optogenetic stimulation enhanced excitatory synaptic transmission in projecting neurons and interneurons of dorsal horn in a manner independent of the rostral ventromedial medulla (RVM).

Anatomical and optogenetic evidence is provided supporting a direct projection from ACC to the dorsal horn neurons and related synapses. The pain-facilitating effect of ACC activation was shown both in control conditions and under spared nerve injury-induced pain sensitization.

Date of Ca^{++} imaging in dorsal horn neurons also support their activation by ACC stimulation.

While descending pain modulation from RVM has been well characterized, a direct pain modulation from cortical areas is rare. In this manuscript, the authors have provided multidisciplinary evidence to demonstrate a direct projection from ACC to neurons in the spinal dorsal horn with presumed excitatory synapses. In the revision, additional data have been added to further support the notion of the ACC effect independent of RVM. Particularly, the new optogenetic data provide much more direct and convincing evidence for the direct pain facilitation by ACC. This represents a significant advance on ACC modulation of pain, and indicates that ACC may have more direct and diverse roles in modulation of sensory pain transmission as well as in modulation of pain-related affect and emotion.

There are some minor issues that need to be addressed to further improve the manuscript.

1. The occlusion of the pain-facilitating effect of ACC stimulation in sensitized pain condition does not necessarily mean the involvement or contribution of ACC (the optogenetic inhibition of the ACC projection does). This is because under pain sensitization, the spinal synaptic transmission studied may have been enhanced enough so that further enhancement by any sites including ACC cannot be observed (ceiling effect). The authors should tune-down the significance of the occlusion experiments (e.g., lines 133-134).

Answer: We agreed with this reviewer, and have tune-down the significance of the conclusion. Please check that in the new manuscript (Page 6, end of the 2nd paragraph).

2. Reference should be provided to support the statement "RVM descending projection mainly affect the deep laminae of the SDH" (line 365).

Answer: We have added two references and mark the changed place in the manuscript.